# Formalizing Learning from Language Feedback with Provable Guarantees

**Wanqiao Xu** [1] [*]   **Allen Nie** [2]   **Ruijie Zheng** [3] [†]   **Aditya Modi** [4] [†]   **Adith Swaminathan** [5] [†]   **Ching-An Cheng** [6] [†]

## Abstract

Interactively learning from observation and language feedback is an increasingly studied area driven by the emergence of large language model (LLM) agents. Despite impressive empirical demonstrations, so far a principled framing of these decision problems remains lacking. We formalize the Learning from Language Feedback (LLF) problem, assert sufficient assumptions to enable learning despite latent rewards, and introduce *transfer eluder dimension* as a measure to characterize the hardness of LLF. We formalize the intuition that information in the language feedback governs the learning complexity, and demonstrate cases where learning from rich language feedback can be exponentially faster than learning from reward. We develop a no-regret algorithm, called HELiX, that provably solves LLF problems through sequential interactions, with performance guarantees that scale with the transfer eluder dimension. Across several empirical domains, we show that HELiX performs well even when repeatedly prompting LLMs does not work reliably. Our contributions mark an important step towards designing principled interactive learning algorithms using generic language feedback.

## 1. Introduction

The vision of general intelligence encompasses systems that learn useful information through rich interactions with the world. Large language models (LLMs) have moved us closer to this goal by enabling agents to interpret natural language feedback such as critique (Du et al., 2023; Akyürek et al., 2023a), guidance (Branavan et al., 2012; Harrison

---
[*]Work started during an internship at Microsoft Research. [†]Work done at Microsoft Research. [1]Stanford University [2]Google DeepMind [3]NVIDIA [4]Meta [5]Netflix [6]Google Research. Correspondence to: Wanqiao Xu <wanqiaoxu@stanford.edu>, Allen Nie <allennie@google.com>, Ching-An Cheng <chingan@google.com>.

*Proceedings of the 43rd International Conference on Machine Learning*, Seoul, South Korea. PMLR 306, 2026. Copyright 2026 by the author(s).

et al., 2018; Scheurer et al., 2023; Nie et al., 2023; Fu et al., 2024; Wei et al., 2024; Cheng et al., 2024), or detailed explanations (Andreas et al., 2018; Chen et al., 2024b; Cheng et al., 2023), whose semantic structure can convey far more information than scalar rewards (Sutton and Barto, 2018). For example, feedback such as "the summary is mostly accurate, but it overlooks the main character's motivation." identifies *what* is wrong and *where* to improve, whereas a reward provides only coarse evaluation.

With LLMs' abilities to understand and respond in natural language (Touvron et al., 2023), language feedback can drastically increase learning efficiency and may be cheaper to obtain than carefully engineered reward signals (Krishna et al., 2023). Despite early works on this topic pre-LLM (Gauthier and Mordatch, 2016; Andreas, 2022) and promising recent empirical results in utilizing language feedback for decision-making (Liu et al., 2023; Chen et al., 2024a; Xie et al., 2024), we still lack a rigorous understanding of when and how language feedback enables efficient learning.

We address this gap by introducing a new, formal mathematical framework of Learning from Language Feedback (LLF) for single-turn in-context learning. The LLF paradigm was introduced by Cheng et al. (2023) as an API to benchmark LLM agents' ability to learn from text feedback in lieu of numerical rewards. Despite empirical evidence that constructive feedback can be more effective for LLM agents than expressing rewards in words (Mu et al., 2022; Liu et al., 2024; Zhong et al., 2024; Xie et al., 2024), the complexity and ambiguity of natural language makes the value of feedback hard to quantify. As a result, it remains unclear when LLF is feasible and how its difficulty compares to classical reward-observed bandit settings. Our contributions are three-fold:

**First, we formalize LLF mathematically and prove that its learning complexity is no larger than that of pure reward learning under suitable assumptions.** We leverage notions of hypothesis testing and elimination in machine learning (De Jong et al., 1993; Lehmann and Romano, 2022) to model the joint generation of language feedback and latent rewards. The interaction protocol of LLF (summarized by Fig. 1) resembles a bandit interface, where agents sequentially interact with the environment through language feedback produced by an underlying hypothesis (without an environmental state). Through defining (1) the concept

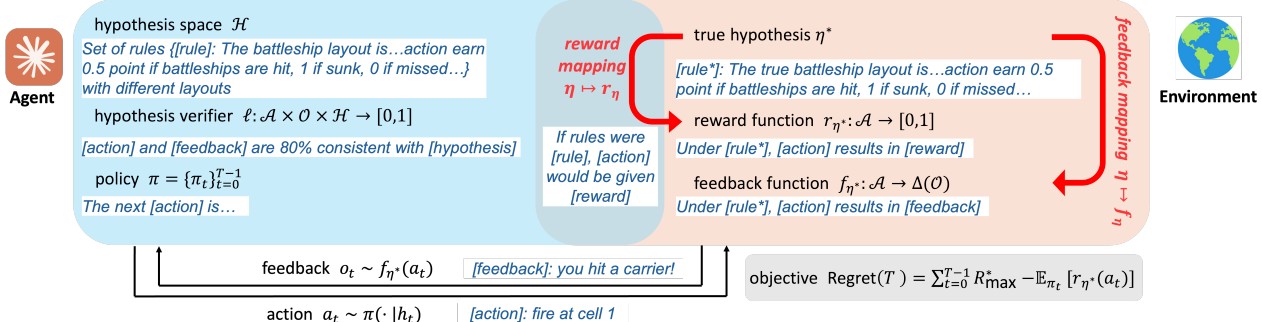

*Figure 1.* An example LLF setup, shown with Battleship game. The environment has a hypothesis $\eta^*$ unknown to the agent. Reward as a function of $\eta^*$ is latent and used only to benchmark the agent. Feedback as a function of $\eta^*$ is observable. No-regret learning is possible, if feedback is *unbiased* (Assumption 3), agent can interpret feedback (Assumption 2) and hypothesis space $\mathcal{H}$ includes $\eta^*$ (Assumption 1).

of a *hypothesis verifier* which processes language feedback, and (2) *transfer eluder dimension* $\dim_{TE}$ which captures information in language feedback, we establish conditions such that LLF is feasible and can be efficiently solved. Under suitable assumptions, we prove that $\dim_{TE} \leq \dim_E$, where $\dim_E$ is the eluder dimension introduced in Russo and Van Roy (2013) for learning from reward.

**Second, we develop a provably no-regret algorithm for LLF with regret guarantees scaling gracefully with learning complexity.** We develop HELiX and prove that over a time horizon $T$, HELiX achieves a regret bound of $\tilde{O}(T^\alpha \dim_{TE}^{1/2})$, where $\alpha \leq 3/4$ depends on properties of the hypothesis verifier. Crucially, our analysis shows that in certain environments, HELiX can be *exponentially* more efficient than learning from reward alone.

**Third, we show that HELiX yields a practical inference-time alternative to common CoT-style prompting.** We show HELiX can be used as an inference-time strategy for LLMs to perform exploration and exploitation, sampling multiple hypotheses and self-verifying as opposed to using a single chain-of-thought (CoT) context. We empirically demonstrate this idea on Battleship and Minesweeper. We show that HELiX yields consistent improvements over standard CoT baselines, providing preliminary evidence that HELiX can be a competitive and principled alternative to CoT for test-time decision-making.

## 2. Learning from Language Feedback

Our first contribution is a formal mathematical model with natural assumptions describing the Learning from Language Feedback (LLF) process (see Fig. 1) so that LLF can be rigorously studied. In what follows, we first define the interaction setup; then we introduce the notion of text hypotheses; finally, we define the hypothesis verifier which evaluates the semantic consistency between candidate hypotheses and observed feedback. These constructions provide a basis for studying LLF's learnability and analyzing regret later.

### 2.1. Formal Setup of LLF

Let $\mathcal{T}$ be a finite set of tokens. We denote the set of all finite token sequences by $\mathcal{T}^+ = \cup_{k \geq 1} \mathcal{T}^k \cup \{\emptyset\}$, where $\mathcal{T}^k$ denotes the set of length-$K$ token sequences. There is a set $\mathcal{O} \subset \mathcal{T}^+$ of token sequences that we refer to as the *feedback* space. For an arbitrary set $\mathcal{X}$, we use $\Delta(\mathcal{X})$ to denote the set of all probability distributions with support on $\mathcal{X}$.

We define the problem of LLF[1] with a finite action set $\mathcal{A}$. At time step $t$, the agent interacts with the environment by executing an action $A_t \in \mathcal{A}$ and observing feedback $O_t \in \mathcal{O}$ sampled from a feedback distribution $f^* : \mathcal{A} \to \Delta(\mathcal{O})$; a reward $R_t = r^*(A_t)$ is incurred, based on a reward function $r^* : \mathcal{A} \to [0, 1]$, though $R_t$ is *not* revealed to the agent. Here we assume the reward is generated by a deterministic function $r^*$; our results can be extended to stochastic rewards. A policy is a distribution on $\mathcal{A}$. We denote the set of policies as $\Pi = \Delta(\mathcal{A})$ and the agent's policy at time step $t$ for sampling $A_t$ as $\pi_t$. We measure the performance of the agent in the LLF setup by regret, which is defined as $\text{Regret}(T) = \sum_{t=0}^{T-1} R_{\max}^* - \mathbb{E}_{\pi_t}[R_t]$, where $T$ is the total number of time steps, $R_{\max}^* = \max_{a \in \mathcal{A}} r^*(a)$, and the expectation is taken over feedback randomness and the algorithm's inner randomization.

This setup is similar to a multi-arm bandit problem, and the goal of the agent is to find actions that maximize the reward. But unlike in bandits, here the agent *does not observe the rewards* $\{R_t\}$, and must learn to maximize the reward solely using natural language feedback $\{O_t\}$. The setup above can be naturally extended to a contextual setting (an analogy of contextual bandit problems; see Appendix E.2 for details).

---

[1]In the LLF protocol (Cheng et al., 2023), a problem context is given before learning to provide background to interpret feedback. We omit writing the problem context but equivalently *assume that the agent can interpret the feedback through the hypothesis verifier.*

## 2.2. Environment Model and Text Hypothesis

The environment in the LLF setup is defined by a feedback function $f^* : \mathcal{A} \rightarrow \Delta(\mathcal{O})$ and a reward function $r^* : \mathcal{A} \rightarrow [0, 1]$. We assume they are "parameterized" by some text description, which we call a *hypothesis*, belonging to a (possibly exponentially large) hypothesis space $\mathcal{H} \subset \mathcal{T}^+$. One can think of a hypothesis as describing the learning problem and mechanism of generating feedback in text such as natural language or code. For example, in a recommendation environment, a hypothesis can be a text description of a user's interests, e.g., "the user enjoys fantasy movies produced in the 21st century..."; in a video game environment, a hypothesis can describe the game's code logic. A hypothesis can also represent a finite-sized numerical array along with operations to decode it into reward and feedback. In short, a hypothesis is a sufficient text description of the learning problem such that the reward and the feedback functions can be fully determined. In our later experiments in Section 5, we will take thinking tokens as a natural form of hypothesis for LLMs.

We model the feedback mechanism through a *feedback mapping* $\eta \mapsto f_\eta$ that maps each hypothesis $\eta \in \mathcal{H}$ to a *feedback function* $f_\eta : \mathcal{A} \rightarrow \Delta(\mathcal{O})$. Similarly, a *reward mapping* $\eta \mapsto r_\eta$ maps a hypothesis $\eta \in \mathcal{H}$ to a *reward function* $r_\eta : \mathcal{A} \rightarrow [0, 1]$. We denote by $\eta^* \in \mathcal{H}$ the true hypothesis of the environment, and use shorthand $f^* = f_{\eta^*}$ and $r^* = r_{\eta^*}$. This construction is reminiscent of classical bandit settings where the reward function is parameterized, such as the linear case $r^*(a) = \phi(a)^\top \theta^*$ for some known feature map $\phi$ and unknown ground-truth parameter $\theta^*$. We generalize this by using the reward mapping $\eta \mapsto r_\eta$ as an analogue of the feature map and the hypothesis $\eta^*$ as the parameter. Following the convention in the literature, we assume that the parameterization, i.e., the reward mapping $\eta \mapsto r_\eta$, is *known* to the agent, but the parameter $\eta^*$ is *unknown*. See Fig. 1 for an overview.

**Assumption 1.** We assume that the agent has access to the reward mapping $r_\eta : \eta \mapsto r_\eta$, where $\eta \in \mathcal{H}$.

The reward mapping can be implemented using an LLM-as-Judge to process a hypothesis text, e.g., to tell whether an action is correct/incorrect (Zheng et al., 2023; Weng et al., 2023; Gu et al., 2024). This assumption can be relaxed to accommodate the agent only knowing an approximated reward mapping, which we discuss in detail in Appendix E.4. We do not assume knowing the feedback mapping $\eta \mapsto f_\eta$, however, as modeling language feedback accurately in practice is difficult.

## 2.3. Measuring Information in Feedback

Without any connection between feedback and reward, learning to minimize regret from feedback is provably impossible.

Intuitively, for LLF to be feasible, language feedback must contain action-relevant information, such as reward, action rankings, or whether an action is optimal. We need a way to quantify this information to study learnability. Since it is impossible to enumerate all possible language feedback, we adopt a weak, implicit definition that exposes only the information the agent can extract.

We introduce the notion of a *hypothesis verifier*, a mechanism that, given an action, observed feedback, and a candidate hypothesis, determines whether the hypothesis is *consistent* with the feedback. This definition is deliberately distinct from a reward model or a correctness oracle. Intuitively, the hypothesis verifier serves a role similar to unit tests. It does not certify that a hypothesis is true, only that it has not been contradicted by what was observed, e.g., a hypothesis verifier may rule out hypotheses that are semantically incompatible with the feedback.

**Assumption 2** (Hypothesis Verifier). There is a hypothesis verifier, which defines a loss $\ell : \mathcal{A} \times \mathcal{O} \times \mathcal{H} \rightarrow [0, 1]$, and the agent has access to the verifier through $\ell$. For any action $a \in \mathcal{A}$, feedback $o \in \mathcal{O}$ and hypothesis $\eta \in \mathcal{H}$, the value $\ell(a, o, \eta)$ quantifies how well $\eta$ aligns with the feedback on action $a$. If $\eta$ is consistent with $o$ on action $a$, then $\ell(a, o, \eta) = 0$; otherwise, it returns a non-zero penalty.

A concrete example may help clarify how a hypothesis verifier loss differs from the way "verifiers" are often used in ML (as a reward signal or directly checking for correctness). Consider a coin-tossing game with unknown head probability $\eta \in \{0, 0.1, 0.5\}$; the agent chooses $a \in \{H, T\}$; the environment returns text feedback $o \in \{\text{"match"}, \text{"miss"}\}$, where "match" means the realized outcome equals the agent's action; latent reward is $r \in \{0, 1\}$, with $r = 1$ iff the agent's choice is correct. Define the hypothesis verifier loss $\ell(a, o, \eta) = \mathbb{1}[(a, o)$ has zero probability under $\eta]$. Given action $a = H$, feedback $o = \text{"match"}$, under hypothesis $\eta = 0$, the hypothesis verifier loss $\ell(a, o, \eta) = 1$ since the action-feedback pair is impossible; However, under a different hypothesis $\eta' = 0.5$, $\ell(a, o, \eta') = 0$ as $\eta'$ is possible. In this example, both reward and feedback contain 1 bit of information, but the distinction is that the verifier loss is an interface for extracting hypothesis-relevant information from language: if we enriched the feedback from "match/miss" to something like "miss—your guess was too optimistic; the coin seems tail-heavy", then $o$ could convey substantially more information than a binary reward, and a suitable $\ell(a, o, \eta)$ can exploit the extra information to eliminate hypotheses more efficiently.

The set of feedback-consistent hypotheses naturally captures information in the feedback. Ideally, feedback generated from $f_\eta(\cdot)$ should be self-consistent, i.e., $\mathbb{E}_{O \sim f_\eta(a)}[\ell(a, O, \eta)] = 0$ for all $a \in \mathcal{A}$ and $\eta \in \mathcal{H}$. However, in practice, both the feedback and the hypothesis veri-

fier may be noisy or imperfect and there may be some $a \in \mathcal{A}$ such that $\mathbb{E}_{O \sim f^*(a)}[\ell(a, O, \eta^*)] > 0$. To accommodate this potential noise while preserving learnability, we adopt a weaker assumption than self-consistency: although the feedback may be noisy, it is *unbiased* such that each hypothesis minimizes the expected hypothesis verifier loss under its induced distribution.

**Assumption 3** (Unbiased Feedback)**.** We say $f_\eta$ is unbiased, if for all $a \in \mathcal{A}$ and $\eta \in \mathcal{H}$, $\eta \in \arg\min_{\eta' \in \mathcal{H}} \mathbb{E}_{O \sim f_\eta(a)}[\ell(a, O, \eta')]$.

This definition can be robustly relaxed to accommodate small approximation errors in observed hypothesis verifier loss, which we discuss in detail in Appendix E.4. The notion of hypothesis verifier can be used to formalize *semantic equivalence* among hypotheses. In natural language, many token sequences share the same underlying semantic meaning. For LLF, such distinctions are not meaningful and should not affect the learning outcome. This invariance can be captured by the hypothesis verifier introduced above. We deem hypotheses as equivalent whenever they induce identical loss functions across all inputs. We use this to define the geometry of the hypothesis space.

**Definition 1** (Hypothesis Equivalence)**.** We define the distance between two hypotheses $\eta, \eta' \in \mathcal{H}$ as $d_\mathcal{H}(\eta, \eta') :=$ $\sup_{a \in \mathcal{A}, o \in \mathcal{O}} |\ell(a, o, \eta) - \ell(a, o, \eta')|$. If $d_\mathcal{H}(\eta, \eta') = 0$, we say $\eta$ and $\eta'$ are *equivalent*.

This definition provides a criterion to determine the equivalence of hypotheses, as two hypotheses with zero distance are indistinguishable from the agent's perspective. The loss function $\ell$ can be designed to reflect semantic similarity, e.g., by assigning similar values to outputs that are paraphrases of one another with embedding-based metrics or LLM-prompted judgments (Wang and Yu, 2023; Chuang et al., 2022; Asai and Hajishirzi, 2020; Bubeck et al., 2023).

**Remark 1.** One may alternatively define a scoring function $g : \mathcal{A} \times \mathcal{O} \to [0, 1]$ that directly evaluates an action-feedback pair and impose some relationships between the scoring function and the underlying reward. This construction is a special case of our framework; see Section 3.3.

## 3. Learnability

Compared to numerical rewards, feedback can potentially carry more information. In LLF, to interpret this feedback and guide learning, the agent is equipped with: *1)* The hypothesis verifier loss function $\ell$ and *2)* The reward mapping $\eta \mapsto r_\eta$. This structure reflects a central feature of LLF: the agent must reason over the hypothesis space $\mathcal{H}$ via the hypothesis verifier to minimize regret of the hidden rewards.

But can an agent learn to maximize reward despite not observing it? For instance, if feedback does not convey useful information for problem solving, it is unrealistic to expect

any learning to happen. On the other hand, if feedback directly reveals the optimal action, then the problem can be solved in two steps. Naturally, one would expect the learnability and complexity of LLF problems to depend on the information that feedback conveys. Here we give natural structures and assumptions to the LLF setup that characterize the difficulty of the learning problem.

### 3.1. Transfer Eluder Dimension

To quantify information in the feedback, we propose a new complexity measure called *transfer eluder dimension* based on the eluder dimension (Russo and Van Roy, 2013) using the hypothesis verifier in Section 2.3. At a high level, transfer eluder dimension characterizes how effectively information in the feedback reduces uncertainty about the unknown reward. When it is small, a single piece of feedback carries a lot of information about the reward, which enables LLF to be more efficient than learning from reward.

**Definition 2.** Given a hypothesis verifier loss $\ell$, an action $a \in \mathcal{A}$ is $\epsilon$-*transfer dependent* on actions $\{a_1, \ldots, a_n\} \subset \mathcal{A}$ with respect to $\mathcal{H}$ if any pair of hypotheses $\eta, \eta' \in \mathcal{H}$ satisfying $\sum_{i=1}^n \left( \mathbb{E}_{O \sim f_{\eta'}(a_i)}[\ell(a_i, O, \eta)] - \ell_{\eta'}^{\min}(a_i) \right) \leq \epsilon^2$, also satisfies $|r_\eta(a) - r_{\eta'}(a)| \leq \epsilon$, where $\ell_\eta^{min}(a) :=$ $\min_{\eta'} \mathbb{E}_{O \sim f_\eta(a)}[\ell(a, O, \eta')]$. Further, $a$ is $\epsilon$-*transfer independent* of $\{a_1, \ldots, a_n\}$ with respect to $\mathcal{H}$ if $a$ is not $\epsilon$-transfer dependent on $\{a_1, \ldots, a_n\}$.

This definition says that an action $a$ is transfer independent of $\{a_1, \ldots, a_n\}$ if two hypotheses, that are both consistent with the feedback on $\{a_1, \ldots, a_n\}$ according to the hypothesis verifier, can differ significantly in their reward predictions at $a$. This differs from the dependency condition used in eluder dimension (Definition 4), which measures discrepancies in both the history and new observation using reward.

**Definition 3** (Transfer eluder dimension)**.** The $\epsilon$-*transfer eluder dimension* $\dim_{TE}(\mathcal{H}, \ell, \epsilon)$ of $\mathcal{H}$ with respect to the hypothesis verifier loss $\ell$ is the length $d$ of the longest sequence of elements in $\mathcal{A}$ such that, for some $\epsilon' \geq \epsilon$, every action element is $\epsilon'$-transfer independent of its predecessors.

Unlike the eluder dimension, transfer eluder dimension measures dependence based on two quantities: the hypothesis verifier loss and the reward function. This extension allows us to capture information in the feedback relevant to reward learning. Later in Section 4, we will present a provable algorithm that attains a sublinear regret rate in LLF in terms of the transfer eluder dimension.

### 3.2. Informative Feedback Reduces Learning Complexity Exponentially

We discuss several example forms of feedback and compute the corresponding transfer eluder dimensions. The nature

*Table 1.* Complexity of LLF w.r.t. different feedback types

| Feedback | $\dim_{TE}(\mathcal{H}, \ell, \epsilon)$ |
|---|---|
| 1. (reward) whether all steps are correct | $O(\|\mathcal{S}\|^L)$ |
| 2. (explanation) index of first incorrect step | $O(\|\mathcal{S}\|L)$ |
| 3. (suggestion) correction of the first mistake | $O(L)$ |
| 4. (demonstration) all the correct steps | $O(1)$ |

of feedback critically affects learning efficiency: uninformative feedback (e.g., random text) leads to infinite transfer eluder dimension, while some feedback provides more information (e.g. about the optimality of selected actions, improving directions, or explanation of mistakes) than reward and accelerates learning. For example, in a constraint satisfaction problem, feedback that reveals satisfied constraints can shrink the set of potentially true hypotheses.

**Example 1** (Reasoning steps). Consider a math reasoning problem where one tries to construct a hidden sequence of $L$-step reasoning $a^* = (s_1^*, \ldots, s_L^*)$, where each $s_i \in \mathcal{S} \subset \mathcal{T}^+$ is a token sequence that represents a correct reasoning at step $i$, and $\mathcal{S}$ is a finite set of token sequences that represent possible reasoning steps. The action set $\mathcal{A} = (\mathcal{T}^+)^L$ consists of all possible reasoning of $L$ steps. Each hypothesis represents a full solution to the problem and rubrics to critique partial answers with. Reward is 1 if all steps are correct and 0 otherwise. Below we show the transfer eluder dimension with $\epsilon < \frac{1}{2L}$ for different feedback (see Appendix C.4 for the exact forms of hypothesis verifiers and proofs). We consider four feedback types, which correspond to the reward, hindsight-negative, hindsight-positive, and future-positive feedback, respectively, in the LLF's feedback taxonomy proposed in (Cheng et al., 2023). Directly learning from rewards incurs exponential complexity, as the agent must enumerate all possible sequences. Feedback that identifies the first mistake enables stage-wise decomposition and yields exponential improvement in $L$, though each stage still requires brute-force search. If the feedback is more constructive, showing not only where the first mistake is but also how to correct it, the problem complexity does not depend on $|\mathcal{S}|$. Finally, if the feedback tells the answer right away, the complexity becomes constant, as the agent can learn the solution immediately after one try.

**Remark 2.** While we could compute the scale of the transfer eluder dimension in the above stylized example, it remains a tool for analysis and will not be known to algorithms we propose in the subsequent sections. In particular, if we're utilizing LLMs to solve LLF, it is not possible to estimate transfer eluder dimension directly.

### 3.3. Learning from Informative Feedback Is No Harder Than Learning from Reward

We have shown examples where the transfer eluder dimension is bounded and decreases as the feedback provides

more information than reward. Here we prove the generality of this observation. We show that if feedback discriminates between rewards, then the transfer eluder dimension of it is no larger than the traditional eluder dimension.

**Definition 4** (Eluder Dimension). An action $a \in \mathcal{A}$ is $\epsilon$-dependent on actions $\{a_1, \ldots, a_n\} \subset \mathcal{A}$ with respect to a reward class $\mathcal{R}$ if any $r, r' \in \mathcal{R}$ satisfying $\sum_{i=1}^n (r(a_i) - r'(a_i))^2 \leq \epsilon^2$, also satisfies $|r(a) - r'(a)| \leq \epsilon$. Further, $a$ is $\epsilon$-independent of $\{a_1, \ldots, a_n\}$ if it is not $\epsilon$-dependent on $\{a_1, \ldots, a_n\}$. The $\epsilon$-eluder dimension $\dim_E(\mathcal{R}, \epsilon)$ of $\mathcal{R}$ is the length $d$ of the longest sequence of elements in $\mathcal{A}$ such that, for some $\epsilon' \geq \epsilon$, every action element is $\epsilon'$-independent of its predecessors.

First, by using the hypothesis verifier, we define the statement "feedback discriminates between rewards".

**Definition 5** (Discriminative feedback). The feedback function $f_\eta$ is *discriminative* of $r_\eta$ with respect to the hypothesis verifier $\ell$ if there is $C_F > 0$ such that $\forall \eta' \in \mathcal{H}$, $a \in \mathcal{A}$, $|r_\eta(a) - r_{\eta'}(a)|^2 \leq C_F \mathbb{E}_{o \sim f_\eta(a)}[\ell(a, o, \eta') - \ell_\eta^{min}(a)]$. We say an LLF problem is *discriminative* if $(f^*, r^*, \ell)$ satisfies the above condition.

This definition states that the hypothesis verifier can distinguish hypotheses based on feedback to the same extent as their reward differences. In other words, if two hypotheses differ in their corresponding rewards, then the hypothesis verifier can tell they are different. Therefore, problems where feedback encodes the reward and hypothesis verifier can decode it (e.g., classical RL) are subsumed as a special case of discriminative LLF. We discuss the relationship of LLF with discriminative feedback and IGL (Xie et al., 2022b) in Appendix A.

A discriminative feedback example is when the unobserved reward is a function of the feedback. Concretely, suppose $r_\eta(a) = \mathbb{E}_{o \sim f_\eta(a)}[g(a, o)]$ for some known $g : \mathcal{A} \times \mathcal{O} \to [0, 1]$. Note that the reward mapping $\eta \mapsto r_\eta$ is known, but the reward function itself is still hidden from the agent (since $\eta^*$ is unknown). Consider $\ell(a, o, \eta) := (g(a, o) - r_\eta(a))^2 = (g(a, o) - \mathbb{E}_{o' \sim f_\eta(a)}[g(a, o')])^2$. Then one can verify that $\eta \in \arg\min_{\eta' \in \mathcal{H}} \mathbb{E}_{o \sim f_\eta(a)}[\ell(a, o, \eta')]$ and show that this feedback-verifier pair is discriminative. (see Appendix C.3). In addition to this example, one can check that the forms of feedback used in Section 3.2 are discriminative too (see Appendix C.4). Discriminative feedback can contain information other than reward as shown in Section 3.2.

With this definition, we show that if feedback can discriminate rewards, the transfer eluder dimension is no larger than the eluder dimension for the reward class induced by $\mathcal{H}$.

**Proposition 1.** *For discriminative LLF problems with $C_F$ as in Definition 5, it holds that*

$$\dim_{TE}(\mathcal{H}, C_F \ell, \epsilon) \leq \dim_E(\mathcal{R}_\mathcal{H}, \epsilon), \text{ where } \mathcal{R}_\mathcal{H} = \{r_\eta : \eta \in \mathcal{H}\} \text{ is the effective reward class of } \mathcal{H}.$$

Proposition 1 implies that discriminative LLF problems are no harder than their reward-only counterparts, for example, running standard UCB over the reward class $\mathcal{R}_{\mathcal{H}}$, where the scalar reward is extracted from language feedback by an LLM. It is important to note that general LLF problems are not necessarily discriminative. This generality is what distinguishes our LLF framework from existing ones such as IGL (Xie et al., 2021): LLF can model settings in which language feedback contains much more *useful* information than reward. For instance, when feedback is not discriminative but reveals information about the optimal action, LLF reflects a reduced learning complexity, whereas a reward-only formulation discards this information and offers no corresponding advantage in IGL.

## 4. HELiX Algorithm

To validate our characterization of learnability based on the transfer eluder dimension, we design a simple UCB-style algorithm, HELiX, outlined in Algorithm 1. HELiX uses feedback to guide exploration following the optimism principle. Given a hypothesis $\eta \in \mathcal{H}$, let $\pi_\eta$ denote its optimal policy. At step $t$, the algorithm maintains a confidence set $\mathcal{H}_t$ of hypotheses that remain approximately consistent with observed actions and feedback. The algorithm then identifies a hypothesis $\eta_o$ that achieves maximal optimistic reward, and follows an optimal policy $\pi_o$ under this hypothesis. With a slight abuse of notation, let $r_\eta(\pi) \coloneqq \sum_{a \in \mathcal{A}} r_\eta(a) \pi(a)$ denote the expected reward of policy $\pi$. An additional design in HELiX compared to standard UCB is a stopping criterion (line 7). It checks for a consensus optimal action among all hypotheses in the confidence set. If the minimax regret $\min_{\pi \in \Pi} \max_{\eta \in \bar{\mathcal{H}}} r_\eta(\pi_\eta) - r_\eta(\pi) = 0$, then the minimizer policy only selects actions that are simultaneously optimal for all candidate hypotheses (see Lemma 5). As discussed in Section 3.3, feedback in a trivial LLF problem can directly reveal the optimal action but nothing about the reward. In this case, the LLF problem is not discriminative, yet the stopping criterion ensures that the algorithm will not over-explore after identifying an optimal action.

### 4.1. Analysis

HELiX is a concrete instantiation of how our conceptual LLF framework can inform algorithmic design, showing that LLF problems with finite transfer eluder dimensions can indeed be solved provably efficiently with a regret guarantee that depends sublinearly on the transfer eluder dimension.

**Theorem 1.** *Under Assumptions 1–3, for all $T \in \mathbb{N}$,* $\mathrm{Regret}(T)$ *of* HELiX *is bounded by*

$$\widetilde{O}\left(T^{3/4}\left(\log N(\mathcal{H}, \epsilon_T^{\mathcal{H}}, d_{\mathcal{H}})\right)^{1/4} \sqrt{\dim_{TE}(\mathcal{H}, \ell, \epsilon_T^{\mathcal{H}})}\right),$$

*where* $N(\mathcal{H}, \epsilon_T^{\mathcal{H}}, d_{\mathcal{H}})$ *denotes the* $\epsilon_T^{\mathcal{H}}$-*covering number of* $\mathcal{H}$ *based on the pseudo-metric* $d_{\mathcal{H}}$, $\dim_{TE}(\mathcal{H}, \ell, \epsilon_T^{\mathcal{H}})$

---

**Algorithm 1** HELiX: Hypothesis Elimination using Language-informed Exploration

1: **Input** $\mathcal{A}, \mathcal{O}, T$, reward mapping $\eta \mapsto r_\eta$, hypothesis verifier loss $\ell : \mathcal{A} \times \mathcal{O} \times \mathcal{H} \to [0, 1]$, confidence levels $\{\epsilon_t\}_{t=0}^{T-1}$
2: **Initialize** $t = 0$, $A_0 \sim \mathrm{Unif}(\mathcal{A})$, $\mathcal{H}_0 = \mathcal{H}$
3: **for** $t = 1, \dots, T$ **do**
4:     observe $O_{t-1}$
5:     $\mathcal{H}_t \leftarrow \mathcal{H}_{t-1} \bigcap \{\eta \in \mathcal{H} : \frac{1}{t} \sum_i \ell(A_i, O_i, \eta) - \min_{\eta' \in \mathcal{H}} \frac{1}{t} \sum_i \ell(A_i, O_i, \eta') \leq \epsilon_t\}$
6:     $(\pi_p, \eta_p) \leftarrow \arg\min_{\pi \in \Pi} \max_{\eta \in \mathcal{H}_t} \left[r_\eta(\pi_\eta) - r_\eta(\pi)\right]$
7:     **if** $r_{\eta_p}(\pi_{\eta_p}) - r_{\eta_p}(\pi_p) = 0$ **then**
8:         $A_t \sim \pi_p(\cdot)$  // Exploitation step: exploit if consensus
9:     **else**
10:         $(\pi_o, \eta_o) \leftarrow \arg\max_{\pi \in \Pi} \max_{\eta \in \mathcal{H}_t} r_\eta(\pi)$  // Exploration step
11:         $A_t \sim \pi_o(\cdot)$
12:     **end if**
13: **end for**

---

*denotes the* $\epsilon_T^{\mathcal{H}}$-*transfer eluder dimension of* $\mathcal{H}$, *and* $\epsilon_T^{\mathcal{H}} = \max\left\{\frac{1}{T^2}, \min_{a \in \mathcal{A}} \inf\{|r_\eta(a) - r^*(a)| : \eta \in \mathcal{H}, \eta \neq \eta^*\}\right\}$.

While the order $\widetilde{O}(T^{3/4})$ on the time horizon $T$ may appear suboptimal compared to classical $\widetilde{O}(\sqrt{T})$ rates for bandit learning with reward feedback, this slower rate is in fact a principled consequence of our minimal assumptions. Up to this point, we make no structural assumptions on the hypothesis verifier loss $\ell$ beyond boundedness. If we have more structural knowledge of $\ell$, say, that it is a squared loss, then the bound can be tightened to match the order $\widetilde{O}(\sqrt{T})$ (see Theorem 3 in Appendix B.4). We provide the proof of Theorem 1 in Appendix B.1.

Directly querying an LLM for an action by prompting with the interaction history (as in Chain of Thought (CoT) reasoning with history) is similar to drawing actions from $\pi_\eta$ where $\eta$ is randomly sampled from $\arg\min_{\eta' \in \mathcal{H}} \sum_i \ell(A_i, O_i, \eta')$. In the RL setting, such a greedy algorithm does not necessarily explore and therefore does not always have low-regret. Since RL is a special case of discriminative LLF, we conjecture that this greedy algorithm also does not have regret guarantees for general LLF. In our experiments, HELiX reliably outperforms this baseline.

### 4.2. Implementing HELiX with Large Language Models

A key contribution of this paper is a practical implementation of HELiX using an LLM. This leads to a novel inference-time exploration strategy distinct from existing approaches to sequential decision-making with LLMs. The latter often rely on CoT-style design and LLMs' inner capabilities to balance exploration and exploitation, which as we will show in experiments can be insufficient even for simple language-based tasks. To achieve principled learning based on hypothesis elimination, our practical implementation of

Algorithm 1 relies on LLMs to approximate three main steps in the theoretical version: (1) maintaining feedback-consistent hypothesis set; (2) minimax exploitation step; (3) UCB-inspired exploration step. The three approximations combined implement Algorithm 1 in a compute-efficient manner. We discuss computational cost in Appendix F.6.

**Approximation of feedback-consistent hypothesis set.** HELiX (Algorithm 1, line 5) maintains a hypothesis set $\mathcal{H}_t$ at iteration $t$ that contains all hypotheses $\eta$ consistent with observed feedback, which will be used to compute $\pi_p$ and $\pi_o$. We approximate searching over the hypothesis set by prompting an LLM to sample $N$ streams of thinking tokens as hypotheses that are consistent with observed feedback as an approximation of $\mathcal{H}_t$ in line 5. When prompted to generate feedback-consistent hypotheses, the LLM implicitly implements a hypothesis verifier, effectively filtering out those that would yield a high hypothesis verifier loss. We chose this design to reduce the number of LLM calls. An alternative design that explicitly prompts the LLM to produce hypothesis verifier loss values requires additional LLM calls and usually has no effect on filtering hypotheses already consistent with feedback as generated above.

**Approximate minimax exploitation step.** HELiX computes a minimax policy $\pi_p$ to check for a consensus optimal action (Algorithm 1, line 7). We approximate this step by searching over finitely many deterministic policies (i.e. actions). We obtain these candidates by prompting an LLM to generate a corresponding optimal action for each hypothesis above (therefore $N$ in total). Then, we construct a score matrix $S_t \in [0,1]^{N \times N}$ whose entries $[S_t]_{\eta,a}$ correspond to the reward of hypothesis-action pairs $(\eta, a)$. To fill $S_t$, we prompt a (possibly different) LLM to score each action above conditioned on a sampled hypothesis. This LLM can be viewed as a reward mapping $r_\eta$ used in line 6. Then, an approximate consensus action $a^*$ is determined as the one that satisfies $[S_t]_{\eta,a^*} \geq [S_t]_{\eta,a}$ for all generated $\eta$.

**Approximate optimistic exploration step.** Using the same $S_t$, we approximate the exploration step in HELiX (Algorithm 1, lines 9-11) by selecting an action that achieves the highest score in $S_t$. This implements exploration using the optimism in the face of uncertainty principle (Auer et al., 2002). We discuss variants and tie-breaking rules in Appendix F.4.

## 5. Experiments

We validate HELiX in two natural language tasks (BATTLESHIP, and MINESWEEPER) in (Tajwar et al., 2025) that require learning from language feedback. Please see Appendix F.1 for the description for each problem. We consider the following agents for comparison.

**CoT.** This agent generates one hypothesis and one action,

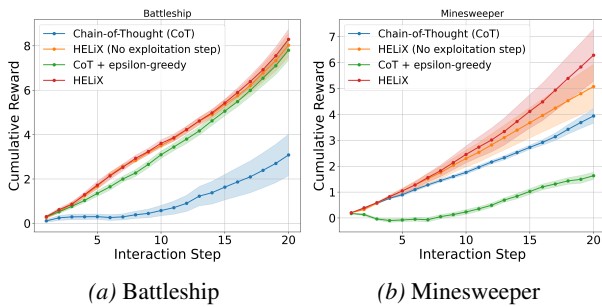

*(a)* Battleship  *(b)* Minesweeper

*Figure 2.* We show the cumulative reward that the agent is able to obtain during a fixed number of interactions with the environment. Shaded area represents the standard error of cumulative reward across different scenarios. We use Claude-Sonnet-3.5 v2.

and returns that action immediately.

**HELiX and HELiX (No exploitation step).** In addition to HELiX, we implement an ablation agent that conducts optimistic exploration without the consensus-based exploitation step (line 7). We demonstrate that optimistic exploration alone is insufficient in our setup.

### 5.1. Discussion of Results

We plot the cumulative reward as a function of the number of environment interaction steps on BATTLESHIP and MINESWEEPER in Figure 2. See a different variant of HELiX in Appendices F.4 and additional studies in F.5. We see that for all problems, the base CoT LLM generally performs worse than HELiX. In environments where information-gathering is essential, such as MINESWEEPER, agents designed to conduct strategic exploration and exploitation tend to outperform the CoT baseline by a large margin. On the other hand, HELiX consistently outperforms both the CoT baseline and the variant HELiX (no exploitation step).

### 5.2. Limitations and Disclaimers

Although the initial results are promising, our practical implementation relies on assumptions that warrant discussion. We assume that the LLM, when prompted, can select an optimal action under a given hypothesis. We also assume that the LLM can produce fair scores across hypotheses for different actions. However, these assumptions may not hold for all LLMs (Shojaee et al., 2025), and further investigation is needed to validate them. Additionally, to capture the agent's uncertainty about the environment, we sample a set of hypotheses from the LLM. These hypotheses should be both diverse and faithful in reflecting the history of interactions. The extent to which existing LLMs can propose plausible hypotheses given historical information remains uncertain, with evidence pointing in both directions (Zhou et al., 2024; Si et al., 2025; Ghareeb et al., 2025). Our theory-inspired algorithm highlights key properties an LLM should exhibit

to function effectively as a decision-making agent, one that autonomously learns from environment feedback, proposes hypotheses, and explores accordingly. Further research is needed to verify whether current LLMs possess these properties.

# 6. Related Work

While using LLMs for general problem solving has been studied for a long time (Xie et al., 2022a; Guo et al., 2024; Akyürek et al., 2023b), relatively fewer prior works studied LLMs for sequential decision-making. Prior methods for leveraging language feedback largely follow two routes. One directly deploys LLMs as agents, incorporating feedback into subsequent prompts or an external memory buffer (Yao et al., 2023; Brooks et al., 2023; Shinn et al., 2023; Wang et al., 2024; Krishnamurthy et al., 2024; Nie et al., 2024; Xi et al., 2023). The other processes feedback to update model weights via fine-tuning (Chen et al., 2024a; Scheurer et al., 2022; Raparthy et al., 2024; Lee et al., 2023; Qu et al., 2025). More recent work targets exploration more explicitly, such as directly learning exploration strategies through supervised fine-tuning (Nie et al., 2024), preference-based learning (Tajwar et al., 2025), or reinforcement learning (Schmied et al., 2025), or prompting LLMs to mimic a perfect Bayesian learner (Arumugam and Griffiths, 2025). However, these results have been empirical. We aim to provide a formal framework and guarantees for learning from language feedback.

Our framework is closely related to multi-armed (Lai and Robbins, 1985) and contextual bandits (Langford and Zhang, 2007) and the designs of no-regret algorithms (Auer et al., 2002; Thompson, 1933; Russo et al., 2018). In particular, our algorithm follows the principle of optimism under uncertainty utilized by UCB (Auer et al., 2002). A key difference is that our algorithm does not observe rewards. Instead we decode information in the feedback through a hypothesis verifier loss to construct confidence sets. In contrast to our unobserved reward setting, recent UCB-like heuristics for LLM agents typically either treat hypotheses as code specifying an MDP (Tang et al., 2024), and/or assume access to the ground-truth numerical reward (Tang et al., 2024; Murthy et al., 2024; Nie et al., 2024).

A separate line of work uses natural language as an auxiliary signal to improve learning. Early studies showed benefits from textual guidance such as manuals (Branavan et al., 2012). Subsequent approaches leveraged grounded language to shape behavior (Gauthier and Mordatch, 2016), guide exploration (Harrison et al., 2018), or learn from feedback (Andreas et al., 2018). More recently, LDD (Zhong et al., 2024) pre-trains agents on language-annotated demonstrations to learn dynamics, then fine-tunes with RL to improve sample efficiency and generalization. While empiri-

cally effective, these approaches do not provide a unifying framework or theoretical guarantees, which is the focus of our work.

More broadly, beyond scalar rewards, rich feedback models have been studied in bandits with side observations (Wang et al., 2003; Kocák et al., 2014), partial monitoring (Bartók et al., 2014), and preference-based feedback (Fürnkranz et al., 2012). These settings commonly treat feedback as informative only insofar as it encodes the underlying reward, and emphasize accurate reward decoding. For example, IGL (Xie et al., 2021) posits a decoder capable of extracting reward estimates from a rich feedback vector, and treats remaining components as distractions. In contrast, we highlight that language feedback can encode learning signals beyond reward, and identify regimes where LLF is strictly *easier* than reward-based learning.

Finally, to characterize sample complexity in reward-aware RL, Russo and Van Roy (2013) introduce the eluder dimension. We extend this notion beyond reward learning (see Fig. 3), opening a new avenue to understanding agent learning in the era of generative AI.

# 7. Conclusion

In this paper, we develop a formal foundation for learning from language feedback (LLF), where agents learn from language feedback rather than scalar rewards. We introduce the transfer eluder dimension as a complexity measure capturing how feedback shapes learning efficiency and show that sufficiently informative feedback can yield exponential gains over reward-based learning. To demonstrate the practicality of this framework, we propose HELiX, a no-regret algorithm with performance guarantees in terms of the transfer eluder dimension.

**Limitations and Open Questions** One might wonder if the transfer eluder dimension forms a lower bound for LLF. The answer, however, is negative, as some LLF instances are trivial despite having infinite transfer eluder dimension. For instance, our LLF framework does not preclude arbitrary rewards while feedback directly reveals an optimal action. The transfer eluder dimension is unbounded in this case, yet learning is easy (and HELiX solves it in one step).

Compared to the demonstration case in Example 1, where binary rewards and a unique optimal action keep the transfer eluder dimension finite, this example exposes a mismatch between worst-case analysis and practice. In particular, our guarantees assume worst-case hypothesis verifier behavior, whereas LLMs impose inductive biases in how they interpret feedback. Empirically, when an optimal action is stated explicitly, LLMs often trust it and act without inferring the full reward structure. HELiX captures this via the early

stopping criterion (line 7), whereas näive reward-driven UCB fails. This counterexample points to a gap in our current understanding: the intrinsic complexity of LLF may lie between worst-case reward identification and optimal behavior learning. Closing this gap by refining the transfer eluder dimension into a regret lower bound remains an important open question. An exciting direction is to introduce an information-theoretic complexity measure capturing the information between the optimal action and observed feedback (Lattimore and Szepesvári, 2019).

Finally, in our experiments, hypotheses are LLM-generated thoughts about the task from reasoning traces, but in general, the hypothesis class is a design choice. Different hypothesis designs can be viewed as different learning targets (Lu et al., 2023), potentially changing both behavior and complexity; characterizing these tradeoffs is an important theoretical and empirical direction.

## Acknowledgements

We thank Dilip Arumugam, the EXAIT @ ICML workshop reviewers, the Netflix Machine Learning and Inference Research group, and the MSR CORE group for helpful feedback and insightful comments on early drafts of this paper. We gratefully acknowledge Benjamin Van Roy for his support in allowing this work to continue beyond the internship.

## Impact Statement

This paper presents work whose goal is to advance the field of Machine Learning. There are many potential societal consequences of our work, none that we feel must be specifically highlighted here.

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

*Figure 3.* **LLF and its relationship to existing paradigms**. LLF covers many existing paradigms: (1) reinforcement learning (RL): agent learning from a scalar reward signal, (2) interaction-grounded learning (IGL) (Xie et al., 2021): agent observes a generic feedback vector that can decode a latent reward signal, (3) discriminative LLF: agent observes language feedback that discriminates between rewards, (4) multi-objective RL: extension of RL to problems with multiple objectives, combined via a utility function, (5) preference-based RL: feedback provides a comparison between two actions, (6) imitation learning: feedback provides expert demonstrations.

## A. LLF and its relationship to existing paradigms

To better understand the position of LLF among existing paradigms of learning from feedback, we provide an in-depth review in this section, as illustrated in Fig. 3. In all discussed paradigms, we focus our comparison on how different forms of feedback are subsumed within LLF, while other environment parameters are loosely assumed to be included in the LLF agent's hypothesis space. LLF covers the following learning paradigms commonly discussed in the literature:

*Table 2.* Comparison of different learning frameworks and their feedback signals. All these learning paradigms are subsumed under the LLF framework with the last column specifying possible verifier losses for an LLF agent.

| Learning Framework | Feedback Type | Discriminative? | LLF Verifier |
|---|---|---|---|
| Reinforcement Learning | Reward $r_{\eta^*} \in \mathbb{R}$ | Yes | $\ell(r_\eta(a), r_{\eta^*}(a))$ |
| Multi-objective RL | Reward vector $r_{\eta^*} \in \mathbb{R}^d$ | Yes | $\ell(r_\eta(a), r_{\eta^*}(a))$ |
| Interaction-Grounded Learning (IGL) | Rich feedback $y \in \mathcal{Y}$ s.t. $\exists \psi^* : \mathcal{Y} \times \mathcal{A} \to \mathbb{R} \approx r_{\eta^*}$ | Yes | Consistency loss: $\ell(y, a, \eta)$ (modeling $\psi^*$ is optional) |
| Preference-based RL | Comparison: $a_1 \overset{\eta^*}{\succ} a_2$ | No | $\mathbb{I}\left[ a_1 \overset{\eta}{\succ} a_2 \right]$ |
| Imitation Learning | Expert actions $a \in \mathcal{A}^*_{\eta^*}$ | No | $\mathbb{I}\left[ a \in \mathcal{A}^*_\eta \right]$ |

**Reinforcement learning (RL)** In RL, upon seeing an environment state $x_t \in \mathcal{X}$, the agent chooses an action $a_t \in \mathcal{A}$ and observes a scalar reward feedback $r_t \in \mathbb{R}$. The rewards and states observed by the agent at any decision step $t$, can depend on the past observed states and actions. In LLF, the agent's hypothesis $\eta \in \mathcal{H}$ returns a reward function $r_\eta : \mathcal{A} \times \mathcal{X} \to [0, 1]$, while the feedback function is exactly the same: $f_\eta = r_\eta$. Hence, RL is trivially subsumed by LLF.

**Partial Monitoring Games** In Partial Monitoring (Bartók et al., 2014), the agent observes an abstract feedback signal (not necessarily reward for its chosen action) and must deduce reward-optimal actions indirectly. The function that maps actions to feedbacks (signal function) is assumed known to the agent, and the challenge is to explore and infer optimal actions indirectly by leveraging the known signal function. In contrast, LLF assumes that the feedback function is unknown, and agents must interpret natural language feedback through a hypothesis verifier to ascertain semantic consistency with hypotheses. The unknown feedback mapping in LLF fundamentally alters the learning challenge, requiring ways to extract insights from potentially ambiguous language feedback, and thus capturing a broader class of interactive learning scenarios.

**Interaction-grounded Learning (IGL)** In IGL (Xie et al., 2021), the environment generates a latent scalar reward $r(x, a) \in [0, 1]$ but only reveals a rich feedback vector $y \in \mathcal{Y}$. To enable learning, the IGL framework assumes reward decodability, i.e., the existence of a decoder $\psi \in \Psi$, such that $\psi : \mathcal{Y} \times \mathcal{A} \to [0, 1]$, capable of extracting reward estimates for the agent. LLF naturally accommodates this by modeling both the latent reward $r_\eta$ and the feedback mapping $f_\eta$ (hence the feedback $y$), allowing the agent to reason about the consistency between the decoded rewards and the observed feedback vectors without needing to identify the true decoder $\psi^*$ or the true feedback function $f^*$.

**Discriminative LLF**  Discriminative LLF, defined formally in Definition 5, subsumes the special case where the latent reward function is itself a function of the observed feedback (Xie et al., 2024). This framework generalizes both RL and IGL, capturing scenarios where feedback is rich and structured (e.g., language) but ultimately reflects reward. As discussed in Section 3.3, this class of LLF problems can be no harder than the reward-only setting and may even improve sample efficiency by leveraging structure in the feedback to recover the reward signal more effectively.

**Multi-objective RL (MORL)**  MORL extends the standard RL framework to environments that return vector-valued rewards rather than a single scalar. The central challenge in MORL is balancing trade-offs across multiple objectives, often handled via scalarization methods (see single-policy learning approaches in (Roijers et al., 2013; Zhang and Golovin, 2020)) or Pareto front exploration (Mossalam et al., 2016). In LLF, this is naturally captured by allowing the agent's hypothesis to represent vector-valued reward functions. Furthermore, the hypothesis verifier loss $\ell : \mathcal{A} \times \mathcal{O} \times \mathcal{H}$ can be extended accordingly. Since the reward vector may be under-determined with respect to the underlying utility function, we treat MORL as distinct from discriminative LLF (Definition 5), which assumes informativeness of feedback with respect to scalar reward.

**Preference-based RL**  In PbRL, the environment does not reveal scalar reward feedback. Instead, the agent receives pairwise preferences over actions (or trajectories), e.g., that action $a$ is preferred over action $a'$. These comparisons may be between actions selected by the agent or between one agent-chosen action and a reference provided by the environment. LLF captures this setting by modeling the feedback function $f_\eta$ as a binary comparator over pairs of actions such that $f_\eta(a, a') \in \{0, 1\}$ indicates the binary preference. The underlying reward model can be implicitly defined in the hypothesis $\eta$ such that it induces such preferences. Thus, this preference-based structure fits within LLF.

**Imitation learning (IL)**  In IL, the agent learns from demonstrations of expert behavior rather than explicit feedback or rewards. To make a closer comparison with LLF, we can consider the interactive imitation learning setting, where the agent observes expert actions (corrections) for all environment observations. IL can be modeled within the LLF framework by considering expert actions as a form of feedback $f_\eta^* = a^*$. Any hypothesis $\eta \in \mathcal{H}$ considered by the LLF agent can evaluate a hypothesis verifier loss which corresponds to the discrepancy between the optimal action of the hypothesis $a_\eta^*$ and expert action $a^*$. IL is thus a special case of LLF where the feedback space is the action space itself, and consistency between the agent's output and expert-labeled actions is the hypothesis verifier loss.

## B. Regret Analysis

### B.1. Proof Sketch

We sketch the proof of the general argument in Theorem 1 in four main steps. The full proof is presented in Appendix B.

**Step 1: Define confidence sets**  For each hypothesis $\eta \in \mathcal{H}$, we define the cumulative population prediction error $\mathcal{L}_t(\eta) = \sum_{i=0}^{t-1} \left( \mathbb{E}_{O \sim f_{\eta^*}(A_i)}[\ell(A_i, O, \eta)] - \ell_{\eta^*}^{\min}(A_i) \right)$ and the cumulative empirical hypothesis verifier loss $L_t(\eta) = \sum_{i=0}^{t-1} \ell(A_i, O_i, \eta) = \sum_{i=0}^{t-1} \ell_i(\eta)$. We define confidence sets $\mathcal{H}_t = \{\eta \in \mathcal{H} : L_t(\eta) \leq \min_{\eta' \in \mathcal{H}} L_t(\eta') + \beta_t\}$ where $\beta_t$ is a confidence parameter.

**Step 2: Regret decomposition**  We let the width of a subset $\mathcal{V} \subseteq \mathcal{H}$ at an action $a \in \mathcal{A}$ be $w_\mathcal{V}(a) = \sup_{\overline{\eta} \in \mathcal{V}} |r_{\overline{\eta}}(a) - r^*(a)|$. Then, we can decompose the regret in terms of version space widths: $\text{Regret}(T, \eta^*) \leq \sum_{t=0}^{T-1} \mathbb{E} \left[ w_{\mathcal{V}_t}(A_t) \cdot \mathbb{1}\{\eta^* \in \mathcal{V}_t\} + \mathbb{1}\{\eta^* \notin \mathcal{V}_t\} \right]$.

**Step 3: Bounding the sum of widths via transfer eluder dimension**  The key step is to show that if the width $w_{\mathcal{H}_t}(A_t) > \epsilon$ for some $\epsilon > 0$, then $A_t$ must be $\epsilon$-dependent on only $O(\beta_t/\epsilon^2)$ disjoint historical action sequences, where $\beta_t$ is the confidence parameter. By the definition of the transfer eluder dimension $d_{TE} = \dim_{TE}(\mathcal{H}, \ell, \epsilon)$, in any sequence of $N$ actions, there must be some action that is $\epsilon$-dependent on at least $\Omega(N/d)$ previous ones. Combining these facts forces the number of large-width version spaces $\sum_{t=0}^{T-1} \mathbb{1}\{w_{\mathcal{H}_t}(A_t) > \epsilon\}$ to be bounded by $O(\beta_T d/\epsilon^2)$. Rearranging terms and choosing a suitable sequence of $\epsilon$ gives that with high probability, $\sum_{t=0}^{T-1} w_{\mathcal{V}_t}(A_t) \leq O(d_{TE} + 2\sqrt{3d_{TE}\beta_T T})$. Note that when the exploitation step is triggered, the per-step regret of all following steps become zero (Lemma 6), and so the regret of HELiX is always bounded above by that without the exploitation step.

**Step 4: Prove high-probability confidence set concentration**  It remains to define suitable $\beta_t$'s and show that $\eta^* \in \mathcal{V}_t$ for

all $t \in \mathbb{N}$ with high probability. Depending on what structural assumptions are known for the hypothesis verifier loss $\ell$, we determine the rate of decay of $\beta_t$. If we only make the minimal assumption that $\ell$ is bounded, then $\beta_T = \widetilde{O}(\sqrt{T})$. Putting everything together proves Theorem 1.

## B.2. Complete Analysis

We first define the version spaces used in the algorithm. As shorthand notations, define

$$\mathcal{L}_t(\eta) = \sum_{i=0}^{t-1} \left( \mathbb{E}_{O \sim f_{\eta^*}(A_i)}[\ell(A_i, O, \eta)] - \ell_{\eta^*}^{\min}(A_i) \right)$$

to be the cumulative population prediction error and

$$L_t(\eta) = \sum_{i=0}^{t-1} \ell(A_i, O_i, \eta) = \sum_{i=0}^{t-1} \ell_i(\eta)$$

to be the cumulative empirical hypothesis verifier loss. A small value of $L_t(\eta)$ means $\eta$ is close to consistent with observed feedback. Let $\mathcal{V}_t \subseteq \mathcal{H}$ be the version space of all hypotheses still plausible after $t$ rounds of interactions. Concretely,

$$\mathcal{V}_t = \{\eta \in \mathcal{H} : L_t(\eta) \leq \min_{\eta' \in \mathcal{H}} L_t(\eta') + \beta_t\}, \tag{1}$$

where $\beta_t > 0$ is an appropriately chosen confidence parameter so that we do not throw away the true hypothesis $\eta^*$ due to noise.

A useful approach to bounding the regret is to decompose it in terms of version spaces. We define the width of a subset $\mathcal{V} \subseteq \mathcal{H}$ at an action $a \in \mathcal{A}$ by

$$w_{\mathcal{V}}(a) = \sup_{\overline{\eta} \in \mathcal{V}} |r_{\overline{\eta}}(a) - r^*(a)|.$$

**Proposition 2** (Regret decomposition). *Fix any sequence $\{\mathcal{V}_t : t \in \mathbb{N}\}$, where $\mathcal{V}_t \subseteq \mathcal{H}$ is measurable with respect to $\sigma(H_t)$. Then for any $T \in \mathbb{N}$,*

$$\text{Regret}(T, \eta^*) \leq \sum_{t=0}^{T-1} \mathbb{E}\left[w_{\mathcal{V}_t}(A_t) \cdot \mathbb{1}\{\eta^* \in \mathcal{V}_t\} + \mathbb{1}\{\eta^* \notin \mathcal{V}_t\}\right].$$

*Proof.* We define the upper bound $U_t(a) = \sup\{r_\eta(a) : \eta \in \mathcal{V}_t\}$ and let $a^* \in \arg\max_{a \in \mathcal{A}} r^*(a)$. When $\eta^* \in \mathcal{V}_t$, the bound $r^*(a) \leq U_t(a)$ hold for all actions. This implies

$$r^*(\eta^*) - r^*(A_t) \leq (U_t(a^*) - r^*(A_t)) \cdot \mathbb{1}\{\eta^* \in \mathcal{V}_t\} + \mathbb{1}\{\eta^* \notin \mathcal{V}_t\}$$
$$\leq w_{\mathcal{V}_t}(A_t) \cdot \mathbb{1}\{\eta^* \in \mathcal{V}_t\} + \mathbb{1}\{\eta^* \notin \mathcal{V}_t\} + [U_t(a^*) - U_t(A_t)] \cdot \mathbb{1}\{\eta^* \in \mathcal{V}_t\}.$$

Since the algorithm selects an action $A_t$ that maximizes $U_t(a)$, the conclusion follows by taking the expectation and summing over all $t = 0, \ldots, T-1$. $\qquad \square$

This proposition reduces upper bounding the regret to bounding the expected sum of widths $\sum_{t=0}^{T-1} \mathbb{E}[w_{\mathcal{V}_t}(A_t)]$ if the version spaces $\mathcal{V}_t$ are constructed such that they contain $\eta^*$ with high probability.

We first introduce a class of Martingale exponential inequalities that will be useful throughout our analysis, including bounding the sum of widths and proving the high-confidence events $\eta^* \in \mathcal{V}_t$. For random variables $(X_t | t \in \mathbb{N})$ adapted to the filtration $(\mathcal{F}_t | t \in \mathbb{N})$, let us assume that $\mathbb{E}[\exp(\lambda X_t)]$ is finite for all $\lambda$ and $\mathbb{E}[X_t | \mathcal{F}_{t-1}] = 0$. We assume that there is a uniform upper bound on the cumulant generating function (i.e., log moment generating function) for the conditional distribution of $X_t$.

**Lemma 1** (Cumulant generating function). *If there is a sequence of convex functions $\{\psi_t : [0, \infty) \to \mathbb{R}\}_{t=0}^{\infty}$ with $\psi_t(0) = 0$ such that, for all $t \in \mathbb{N}$ and all $\lambda \in [0, \infty)$,*

$$\log \mathbb{E}\left[e^{\lambda |X_t|} | \mathcal{F}_{t-1}\right] \leq \psi_t(\lambda),$$

*then for all $\delta \in (0,1)$ and $T \in \mathbb{N}$, with probability $1 - \delta$,*

$$\left| \sum_{t=0}^{T-1} X_t \right| \leq \inf_{\lambda \in [0,\infty)} \left\{ \frac{\sum_{t=0}^{T-1} \psi_t(\lambda) + \log(2/\delta)}{\lambda} \right\}.$$

*Proof.* Let $S_T = \sum_{t=0}^{T-1} X_t$. By Markov's inequality, for all $u \in \mathbb{R}$ and $\lambda \in [0,\infty)$,

$$\mathbb{P}\left(S_T \geq u\right) = \mathbb{P}\left(e^{\lambda S_T} \geq e^{\lambda u}\right) \leq \frac{\mathbb{E}[e^{\lambda S_T}]}{e^{\lambda u}} = \frac{\mathbb{E}[\mathbb{E}[e^{\lambda S_T} | \mathcal{F}_{T-1}]]}{e^{\lambda u}} = \frac{\mathbb{E}[e^{\lambda \sum_{t=0}^{T-2} X_t} \mathbb{E}[e^{\lambda X_{T-1}} | \mathcal{F}_{T-1}]]}{e^{\lambda u}}$$

$$\leq \frac{\mathbb{E}[e^{\lambda \sum_{t=0}^{T-2} X_t}] \exp(\psi_{T-1}(\lambda))}{e^{\lambda u}} \leq \cdots \leq \frac{\exp(\sum_{t=0}^{T-1} \psi_t(\lambda))}{e^{\lambda u}}.$$

This gives

$$\mathbb{P}\left(S_T \geq u\right) \leq \exp\left(-\lambda u + \sum_{t=0}^{T-1} \psi_t(\lambda)\right)$$

for all $\lambda \in [0,\infty)$. Applying the same argument to $-X_t$, we have

$$\mathbb{P}\left(S_T \leq -u\right) = \mathbb{P}\left(-S_T \geq u\right) \leq \exp\left(-\lambda u + \sum_{t=0}^{T-1} \psi_t(\lambda)\right).$$

Solving for $u$ to achieve a $\delta/2$ probability for each side, and taking the infimum over $\lambda \in [0,\infty)$, we have with probability at least $1 - \delta$,

$$S_T \leq \inf_{\lambda \in [0,\infty)} \left\{ \frac{\sum_{t=0}^{T-1} \psi_t(\lambda) + \log(2/\delta)}{\lambda} \right\}.$$

$\square$

We now proceed to bounding the sum of widths $\sum_{t=0}^{T-1} \mathbb{E}[w_{\mathcal{V}_t}(A_t)]$ when the event $\eta^* \in \mathcal{V}_t$ holds. As a first step, we show that there cannot be many version spaces $\mathcal{V}_t$ with a large width. For all $t \in \mathbb{N}$ and $\eta, \eta' \in \mathcal{H}$, we define the martingale difference

$$Z_t(\eta, \eta') = \mathbb{E}_{O \sim f_{\eta^*}(A_t)}\left[\ell(A_t, O, \eta) - \ell(A_t, O, \eta') | \mathcal{G}_{t-1}\right] - \left(\ell(A_t, O_t, \eta) - \ell(A_t, O_t, \eta')\right).$$

Notice that $Z_t$ have expectation zero and constitutes a martingale difference sequence adapted to the filtration $(\mathcal{G}_t | t \in \mathbb{N})$ where $\mathcal{G}_t$ is the $\sigma$-algebra generated by all observations $\{(a_0, o_1), \ldots, (a_t, o_t)\}$ up to time $t$.

**Proposition 3.** *If the conditions in Lemma 1 holds for $(Z_t | t \in \mathbb{N})$ adapted to $(\mathcal{G}_t | t \in \mathbb{N})$ with cumulative generating function bound $(\psi_t | t \in \mathbb{N})$, $(\beta_t \geq 0 | t \in \mathbb{N})$ in (1) is a nondecreasing sequence such that for all $t \in \mathbb{N}$, $\beta_t \geq \inf_{\lambda \in [0,\infty)} \left\{ \frac{\sum_{i=0}^{t-1} \psi_i(\lambda) + \log(10t^2/3\delta)}{\lambda} \right\}$, then for all $\delta \in (0,1)$, with probability at least $1 - \delta$,*

$$\sum_{t=0}^{T-1} \mathbb{1}\{w_{\mathcal{V}_t}(A_t) > \epsilon\} \cdot \mathbb{1}\{\eta^* \in \mathcal{V}_t\} \leq \left(\frac{3\beta_T}{\epsilon^2} + 1\right) \dim_{TE}(\mathcal{H}, \ell, \epsilon)$$

*for all $T \in \mathbb{N}$ and $\epsilon > 0$.*

*Proof.* We first show that if $w_{\mathcal{V}_t}(A_t) > \epsilon$ and $\eta^* \in \mathcal{V}_t$, then with high probability, $A_t$ is $\epsilon$-dependent on fewer than $O(\beta_t/\epsilon^2)$ disjoint subsequences of $(A_0, A_1, \ldots, A_{t-1})$. If $w_{\mathcal{V}_t}(A_t) > \epsilon$ and $\eta^* \in \mathcal{V}_t$, there exists $\overline{\eta} \in \mathcal{V}_t$ such that $|r_{\overline{\eta}}(A_t) - r_{\eta^*}(A_t)| > \epsilon$. By definition, if $A_t$ is $\epsilon$-dependent on a subsequence $(A_{i_1}, \ldots, A_{i_k})$ of $(A_0, \ldots, A_{t-1})$, then we have that

$$\sum_{j=1}^{k} \left( \mathbb{E}_{O \sim f_{\eta^*}(A_{i_j})}[\ell(A_{i_j}, O, \overline{\eta})] - \ell_{\eta^*}^{\min}(A_{i_j}) \right) > \epsilon^2.$$

It follows that if $A_t$ is $\epsilon$-dependent on $K$ disjoint subsequences of $(A_0, \ldots, A_{t-1})$ then

$$\sum_{i=0}^{t-1} \left( \mathbb{E}_{O \sim f_{\eta^*}(A_i)}[\ell(A_i, O, \overline{\eta})] - \ell_{\eta^*}^{\min}(A_i) \right) > K\epsilon^2.$$

Then

$$\sum_{i=0}^{t-1} \left( \mathbb{E}_{O \sim f_{\eta^*}(A_i)}[\ell(A_i, O, \overline{\eta})] - \ell_{\eta^*}^{\min}(A_i) \right)$$

$$= \sum_{i=0}^{t-1} \mathbb{E}_{O \sim f_{\eta^*}(A_i)} \left[ \ell(A_i, O, \overline{\eta}) - \ell(A_i, O, \eta^*) \right]$$

$$= \left[ \sum_{i=0}^{t-1} \ell(A_i, O_i, \eta^*) - \min_{\eta' \in \mathcal{H}} \sum_{i=0}^{t-1} \ell(A_i, O_i, \eta') \right] - \left[ \sum_{i=0}^{t-1} \ell(A_i, O_i, \overline{\eta}) - \min_{\eta' \in \mathcal{H}} \sum_{i=0}^{t-1} \ell(A_i, O_i, \eta') \right]$$

$$+ \left[ \sum_{i=0}^{t-1} [\ell(A_i, O_i, \overline{\eta}) - \ell(A_i, O_i, \eta^*)] - \sum_{i=0}^{t-1} \mathbb{E}_{O \sim f_{\eta^*}(A_i)} \left[ \ell(A_i, O, \overline{\eta}) - \ell(A_i, O, \eta^*) \right] \right]$$

$$\leq \left| \sum_{i=0}^{t-1} \ell(A_i, O_i, \eta^*) - \min_{\eta' \in \mathcal{H}} \sum_{i=0}^{t-1} \ell(A_i, O_i, \eta') \right| + \left| \sum_{i=0}^{t-1} \ell(A_i, O_i, \overline{\eta}) - \min_{\eta' \in \mathcal{H}} \sum_{i=0}^{t-1} \ell(A_i, O_i, \eta') \right|$$

$$+ \left[ \sum_{i=0}^{t-1} [\ell(A_i, O_i, \overline{\eta}) - \ell(A_i, O_i, \eta^*)] - \sum_{i=0}^{t-1} \mathbb{E}_{O \sim f_{\eta^*}(A_i)} \left[ \ell(A_i, O, \overline{\eta}) - \ell(A_i, O, \eta^*) \right] \right]$$

$$\leq 2\beta_t + \sum_{i=0}^{t-1} [\ell(A_i, O_i, \overline{\eta}) - \ell(A_i, O_i, \eta^*)] - \sum_{i=0}^{t-1} \mathbb{E}_{O \sim f_{\eta^*}(A_i)} \left[ \ell(A_i, O, \overline{\eta}) - \ell(A_i, O, \eta^*) \right]$$

$$= 2\beta_t - \sum_{i=0}^{t-1} Z_i(\overline{\eta}, \eta^*).$$

Using Lemma 1,

$$\mathbb{P} \left( \left| \sum_{i=0}^{t-1} Z_i(\overline{\eta}, \eta^*) \right| > \inf_{\lambda \in [0, \infty)} \left\{ \frac{\sum_{i=0}^{t-1} \psi_i(\lambda) + \log(2/\delta)}{\lambda} \right\} \right) \leq \delta.$$

We choose a sequence $\{\delta_t\}_{t \in \mathbb{N}_{>0}}$ where $\delta_t = \frac{3\delta}{5t^2}$, and so $\sum_{t=1}^{\infty} \delta_t < \delta$. Using a union bound over all $t \in \mathbb{N}_{>0}$, we have that with probability at least $1 - \delta$, for all $t \in \mathbb{N}$,

$$\left| \sum_{i=0}^{t-1} Z_i(\overline{\eta}, \eta^*) \right| \leq \inf_{\lambda \in [0, \infty)} \left\{ \frac{\sum_{i=0}^{t-1} \psi_i(\lambda) + \log(10t^2/3\delta)}{\lambda} \right\} \leq \beta_t.$$

Since $\{\beta_t\}_{t \in \mathbb{N}}$ is nondecreasing in $t$, we have that with probability at least $1 - \delta$, $K\epsilon^2 \leq 3\beta_T$. It then follows that with probability at least $1 - \delta$, $K \leq 3\beta_T/\epsilon^2$.

Next, we take any action sequence $(a_1, \ldots, a_\tau)$ and show that there is some element $a_j$ that is $\epsilon$-dependent on at least $\tau/d - 1$ disjoint subsequences of $(a_1, \ldots, a_{j-1})$, where $d = \dim_{TE}(\mathcal{H}, \ell, \epsilon)$. For an integer $K$ satisfying $Kd + 1 \leq \tau \leq Kd + d$, we will construct $K$ disjoint subsequences $B_1, \ldots, B_K$ inductively starting with $B_i = (a_i)$ for $i = 1, \ldots, K$. If $a_{K+1}$ is $\epsilon$-dependent on each subsequence $B_1, \ldots, B_K$, we are done. Otherwise, there must be at least one subsequence for which $a_{K+1}$ is $\epsilon$-independent. We choose such a subsequence and append $a_{K+1}$ to it. We will repeat this process for $a_j$ with $j = K + 2, K + 3, \ldots$ until either $a_j$ is $\epsilon$-dependent on each subsequence or $j = \tau$. If the first case occurs, we are done. If $j = \tau$, we necessarily have that $\sum |B_i| \geq Kd$. Since each element of a subsequence $B_i$ is $\epsilon$-independent of its predecessors, $|B_i| = d$. By the definition of $\dim_{TE}(\mathcal{H}, \ell, \epsilon)$, $a_\tau$ must be $\epsilon$-dependent on each subsequence.

We now take $(A_1, \ldots, A_\tau)$ to be the subsequence $(A_{t_1}, \ldots, A_{t_\tau})$ of $(A_1, \ldots, A_T)$ where for each $A_t$, we have $w_{\mathcal{V}_t}(A_t) > \epsilon$. As we have shown first, with probability at least $1 - \delta$, each $A_{t_j}$ is $\epsilon$-dependent on fewer than $3\beta_T/\epsilon^2$ disjoint subsequences of $(A_1, \ldots, A_{j-1})$. As we have shown in the preceding paragraph, there is some $a_j$ that is $\epsilon$-dependent on at least $\tau/d - 1$

disjoint subsequences of $(a_1, \ldots, a_{j-1})$. Combining these two facts, we may conclude that $\tau/d - 1 \leq 3\beta_T/\epsilon^2$. It follows that with probability at least $1 - \delta$, $\tau \leq \left(3\beta_T/\epsilon^2 + 1\right) d$ as desired. $\qquad \square$

We are now ready to bound the sum of widths $\sum_{t=0}^{T-1} \mathbb{E}[w_{\mathcal{V}_t}(A_t)]$ when the event $\eta^* \in \mathcal{V}_t$ holds. Consider the $\epsilon_T^{\mathcal{H}}$-transfer eluder dimension of $\mathcal{H}$, where

$$\epsilon_t^{\mathcal{H}} = \max \left\{ \frac{1}{t^2}, \min_{a \in \mathcal{A}} \inf\{|r_\eta(a) - r^*(a)| : \eta \in \mathcal{H}, \eta \neq \eta^*\} \right\}. \tag{2}$$

**Lemma 2.** *If the conditions in Lemma 1 holds for $(Z_t | t \in \mathbb{N})$ adapted to $(\mathcal{G}_t | t \in \mathbb{N})$ with cumulative generating function bound $(\psi_t | t \in \mathbb{N})$, $(\beta_t \geq 0 | t \in \mathbb{N})$ in (1) is a nondecreasing sequence such that for all $t \in \mathbb{N}$, $\beta_t \geq \inf_{\lambda \in [0,\infty)} \left\{ \frac{\sum_{i=0}^{t-1} \psi_i(\lambda) + \log(10t^2/3\delta)}{\lambda} \right\}$, then for all $\delta \in (0, 1)$, with probability at least $1 - \delta$,*

$$\sum_{t=0}^{T-1} w_{\mathcal{V}_t}(A_t) \cdot \mathbb{1}\{\eta^* \in \mathcal{V}_t\} \leq \frac{1}{T} + \min\left\{\dim_{TE}(\mathcal{H}, \ell, \epsilon_T^{\mathcal{H}}), T\right\} + 2\sqrt{3 \dim_{TE}(\mathcal{H}, \ell, \epsilon_T^{\mathcal{H}}) \beta_T T}$$

*for all $T \in \mathbb{N}$.*

*Proof.* Let $d_T = \dim_{TE}(\mathcal{H}, \ell, \epsilon_T^{\mathcal{H}})$ and $w_t = w_{\mathcal{V}_t}(A_t)$. Reorder the sequence $(w_1, \ldots, w_T) \to (w_{i_1}, \ldots, w_{i_T})$ where $w_{i_1} \geq w_{i_2} \geq \cdots \geq w_{i_T}$. We have

$$\sum_{t=0}^{T-1} w_{\mathcal{V}_t}(A_t) \cdot \mathbb{1}\{\eta^* \in \mathcal{V}_t\}$$

$$= \sum_{t=0}^{T-1} w_{i_t} \cdot \mathbb{1}\{\eta^* \in \mathcal{V}_{i_t}\}$$

$$= \sum_{t=0}^{T-1} w_{i_t} \cdot \mathbb{1}\{\eta^* \in \mathcal{V}_{i_t}\} \cdot \mathbb{1}\{w_{i_t} > \epsilon_T^{\mathcal{H}}\} + \sum_{t=0}^{T-1} w_{i_t} \cdot \mathbb{1}\{\eta^* \in \mathcal{V}_{i_t}\} \cdot \mathbb{1}\{w_{i_t} \leq \epsilon_T^{\mathcal{H}}\}$$

$$\leq \frac{1}{T} + \sum_{t=0}^{T-1} w_{i_t} \cdot \mathbb{1}\{\eta^* \in \mathcal{V}_{i_t}\} \cdot \mathbb{1}\{w_{i_t} > \epsilon_T^{\mathcal{H}}\}.$$

The last inequality follows since either $\epsilon_T^{\mathcal{H}} = 1/T^2$ and $\sum_{t=0}^{T-1} \epsilon_T^{\mathcal{H}} = 1/T$ or $\epsilon_T^{\mathcal{H}}$ is set below the smallest possible width and hence $\mathbb{1}\{w_{i_t} \leq \epsilon_T^{\mathcal{H}}\}$ never occurs. We have that $w_{i_t} \leq 1$. Also, $w_{i_t} > \epsilon \iff \sum_{k=0}^{T-1} \mathbb{1}\{w_{\mathcal{V}_k}(a_k) > \epsilon\} \geq t$. By Proposition 3, with probability at least $1 - \delta$, this can only happen if $t < \left(3\beta_T/\epsilon^2 + 1\right) \dim_{TE}(\mathcal{H}, \ell, \epsilon)$. For $\epsilon \geq \epsilon_T^{\mathcal{H}}$, since $\dim_{TE}(\mathcal{H}, \ell, \epsilon')$ is non-increasing in $\epsilon'$, $\dim_{TE}(\mathcal{H}, \ell, \epsilon) \leq \dim_{TE}(\mathcal{H}, \ell, \epsilon_T^{\mathcal{H}}) = d_T$. Therefore, when $w_{i_t} > \epsilon \geq \epsilon_T^{\mathcal{H}}$, $t \leq \left(3\beta_T/\epsilon^2 + 1\right) d_T$, implying $\epsilon \leq \sqrt{\frac{3\beta_T d_T}{t - d_T}}$. So if $w_{i_t} > \epsilon_T^{\mathcal{H}}$, then $w_{i_t} \leq \min\{1, \sqrt{\frac{3\beta_T d_T}{t - d_T}}\}$. Thus,

$$\sum_{t=0}^{T-1} w_{i_t} \cdot \mathbb{1}\{\eta^* \in \mathcal{V}_{i_t}\} \cdot \mathbb{1}\{w_{i_t} > \epsilon_T^{\mathcal{H}}\} \leq d_T + \sum_{t=d_T+1}^{T-1} \sqrt{\frac{3\beta_T d_T}{t - d_T}}$$

$$\leq d_T + \sqrt{3\beta_T d_T} \int_{t=1}^{T-1} \frac{1}{\sqrt{t}} dt$$

$$= d_T + 2\sqrt{3\beta_T d_T T}.$$

Since the sum of widths is always bounded by $T$, this implies that with probability $1 - \delta$,

$$\sum_{t=0}^{T-1} w_{\mathcal{V}_t}(a_t) \cdot \mathbb{1}\{\eta^* \in \mathcal{V}_t\}$$

$$\leq \min\left\{T, \frac{1}{T} + \dim_{TE}(\mathcal{H}, \ell, \epsilon_T^{\mathcal{H}}) + 2\sqrt{3 \dim_{TE}(\mathcal{H}, \ell, \epsilon_T^{\mathcal{H}}) \beta_T T}\right\}$$

$$\leq \frac{1}{T} + \min\left\{\dim_{TE}(\mathcal{H}, \ell, \epsilon_T^{\mathcal{H}}), T\right\} + 2\sqrt{3\dim_{TE}(\mathcal{H}, \ell, \epsilon_T^{\mathcal{H}})\beta_T T}.$$

$\square$

So far, we have only considered HELiX without the exploitation step. We remark that by Lemma 6, when the exploitation step is triggered, the per-step regret of all following steps become zero, and so the regret of the full HELiX is always bounded above by that without the exploitation step. Combining this observation with Lemma 2 and Proposition 2, we arrive at the following abstract regret bound in terms of the version space confidence parameter $\beta_T$.

**Theorem 2.** *If it holds that for some $\delta \in (0, 1)$, with probability at least $1 - \delta$, $\eta^* \in \mathcal{V}_t$ for all $t$, then for all $T \in \mathbb{N}$,*

$$\mathrm{Regret}(T) \leq 1 + \frac{1}{T} + \min\{\dim_{TE}(\mathcal{H}, \ell, \epsilon_T^{\mathcal{H}}), T\} + 2\sqrt{3\dim_{TE}(\mathcal{H}, \ell, \epsilon_T^{\mathcal{H}})\beta_T T}.$$

The dominant term in the regret bound is

$$2\sqrt{3\dim_{TE}(\mathcal{H}, \ell, \epsilon_T^{\mathcal{H}})\beta_T T}.$$

For our main theorem, it remains to design suitable version spaces $\mathcal{V}_t$ and show that they contain the true hypothesis $\eta^*$ with high probability. Crucially, the rate at which the confidence parameters $\beta_t$ of these version spaces shrink depends on concentration properties of the hypothesis verifier loss function $\ell$. Note that for the general LLF framework, we have assumed only that $\ell$ is a bounded function taking values in $[0, 1]$. If we have more structural assumptions on the hypothesis verifier loss $\ell$, for example, that $\ell$ is $\alpha$-strongly convex, then we may arrive at a tighter regret bound up to order $\sqrt{T}$ by taking $\beta_T$ to be of constant order.

### B.3. Version Space Construction for General Bounded Loss

Consider the most general case with minimal assumptions on the loss function, namely, that it is bounded between $[0, 1]$ for all inputs. Then we prove the following high-probability event:

**Lemma 3** (High-probability event). *For all $\delta > 0$, $\eta, \eta' \in \mathcal{H}$,*

$$\mathbb{P}\left(\mathcal{L}_T(\eta') \geq \mathcal{L}_T(\eta) + L_T(\eta') - L_T(\eta) - \sqrt{2T\log\left(\frac{10T^2}{3\delta}\right)}, \quad \forall T \in \mathbb{N}\right) \geq 1 - \delta.$$

*Proof.* For each $t = 1, \ldots, T$, define the Martingale difference sequence

$$X_t = \mathbb{E}_{O \sim f_{\eta^*}(A_t)}\left[\ell(A_t, O, \eta) - \ell(A_t, O, \eta')\right] - (\ell(A_t, O_t, \eta) - \ell(A_t, O_t, \eta')).$$

$$\mathcal{L}_T(\eta') - \mathcal{L}_T(\eta) - (L_T(\eta') - L_T(\eta))$$
$$= \sum_{t=0}^{T-1}\left(\mathbb{E}_{O \sim f_{\eta^*}(A_t)}[\ell(A_t, O, \eta)] - \mathbb{E}_{O \sim f_{\eta^*}(A_t)}[\ell(A_t, O, \eta')]\right) - \sum_{t=0}^{T-1}(\ell(A_t, O_t, \eta) - \ell(A_t, O_t, \eta'))$$
$$= \sum_{t=0}^{T-1}\mathbb{E}_{O \sim f_{\eta^*}(A_t)}\left[\ell(A_t, O, \eta) - \ell(A_t, O, \eta')\right] - \sum_{t=0}^{T-1}(\ell(A_t, O_t, \eta) - \ell(A_t, O_t, \eta'))$$
$$= \sum_{t=0}^{T-1}X_t.$$

Notice that $X_t$ have expectation zero and constitutes a Martingale difference sequence adapted to the filtration $\{\mathcal{G}_t\}_{t \geq 1}$ where $\mathcal{G}_t$ is the $\sigma$-algebra generated by all observations $\{(A_0, O_1), \ldots, (A_t, O_t)\}$ up to time $t$. Since feedback losses $\ell(a, o, \eta)$ are uniformly bounded between $[0, 1]$, we have that $X_t \in [-2, 2]$ with probability 1. Using Lemma 1 with $\psi_t(\lambda) = \lambda^2/2$ and taking the infimum over $\lambda$, we get

$$\mathbb{P}\left(\left|\sum_{t=0}^{T-1}X_t\right| > \sqrt{2T\log(2/\delta)}\right) \leq \delta.$$

We choose a sequence $\{\delta_T\}_{T \in \mathbb{N}_{>0}}$ where $\delta_T = \frac{3\delta}{5T^2}$ such that $\sum_{T=1}^{\infty} \delta_T < \delta$. Using a union bound over all $T \in \mathbb{N}_{\geq 0}$, we have that with probability at least $1 - \delta$,

$$|\mathcal{L}_T(\eta') - \mathcal{L}_T(\eta) - (L_T(\eta') - L_T(\eta))| \leq \sqrt{2T \log\left(\frac{2}{\delta_T}\right)} = \sqrt{2T \log\left(\frac{10T^2}{3\delta}\right)} \quad \forall T \in \mathbb{N}.$$

$\square$

Since $\eta^*$ is the true hypothesis, by Assumption 3, it minimizes the population loss $\mathcal{L}_T(\eta)$ for all $T \in \mathbb{N}$. That is, for all $\eta \in \mathcal{H}$,

$$\mathcal{L}_T(\eta^*) \leq \mathcal{L}_T(\eta) \quad \forall T \in \mathbb{N}.$$

Suppose $m = |\mathcal{H}| < \infty$. By Lemma 3, for any $\eta \in \mathcal{H}$, with probability at least $1 - \delta/m$, for all $T \in \mathbb{N}$,

$$L_T(\eta^*) - L_T(\eta) \leq \mathcal{L}_T(\eta^*) - \mathcal{L}_T(\eta) + \sqrt{2T \log\left(\frac{10T^2}{3\delta}\right)} \leq \sqrt{2T \log\left(\frac{10mT^2}{3\delta}\right)}.$$

Using a union bound over $\mathcal{H}$, with probability at least $1 - \delta$, the true hypothesis $\eta^*$ is contained in the version space

$$\mathcal{V}_T = \left\{ \eta \in \mathcal{H} : L_T(\eta) \leq \min_{\eta' \in \mathcal{H}} L_T(\eta') + \sqrt{2T \log\left(\frac{10|\mathcal{H}|T^2}{3\delta}\right)} \right\}$$

for all $T \in \mathbb{N}$. To extend this to a space of infinite hypotheses, we measure the set $\mathcal{H}$ by some discretization scale $\alpha$. Recall that we define distances in the hypothesis space in terms of the loss function $\ell$:

$$d_{\mathcal{H}}(\eta, \eta') = \sup_{a \in \mathcal{A}, o \in \mathcal{O}} |\ell(a, o, \eta) - \ell(a, o, \eta')|.$$

**Lemma 4.** $d_{\mathcal{H}}(\cdot, \cdot)$ *is a pseudometric on* $\mathcal{H}$.

*Proof.* We check the axioms for a pseudometric.

- nonnegativity: $d_{\mathcal{H}}(\eta, \eta) = 0$ and $d_{\mathcal{H}}(\eta, \eta') \geq 0$ for all $\eta, \eta' \in \mathcal{H}$.

- symmetry: $d_{\mathcal{H}}(\eta, \eta') = d_{\mathcal{H}}(\eta', \eta)$.

- triangle inequality: for each $a \in \mathcal{A}$ and $o \in \mathcal{O}$, $|\ell(a, o, \eta) - \ell(a, o, \eta'')| \leq |\ell(a, o, \eta) - \ell(a, o, \eta')| + |\ell(a, o, \eta') - \ell(a, o, \eta'')|$. Taking the supremum over $\mathcal{A}$ and $\mathcal{O}$ yields the desired property.

$\square$

Let $N(\mathcal{H}, \alpha, d_{\mathcal{H}})$ denote the $\alpha$-covering number of $\mathcal{H}$ in the pseudometric $d_{\mathcal{H}}$, and let

$$\beta_t^*(\mathcal{H}, \delta, \alpha) := \sqrt{2t \log\left(\frac{10N(\mathcal{H}, \alpha, d_{\mathcal{H}})t^2}{3\delta}\right)} + 2\alpha t. \tag{3}$$

**Proposition 4.** *For* $\delta > 0$, $\alpha > 0$, *and* $T \in \mathbb{N}$, *define*

$$\mathcal{V}_T := \left\{ \eta \in \mathcal{H} : L_T(\eta) \leq \min_{\eta' \in \mathcal{H}} L_T(\eta') + \beta_T^* \right\}$$

*Then it holds that*

$$\mathbb{P}\left( \eta^* \in \bigcap_{T=1}^{\infty} \mathcal{V}_T \right) \geq 1 - \delta.$$

*Proof.* Let $\mathcal{H}^\alpha \subseteq \mathcal{H}$ be an $\alpha$-cover of $\mathcal{H}$ in the pseudometric $d_\mathcal{H}$. In other words, for any $\eta \in \mathcal{H}$, there is an $\eta^\alpha \in \mathcal{H}^\alpha$ such that $d_\mathcal{H}(\eta, \eta^\alpha) \leq \alpha$. A union bound over $\mathcal{H}^\alpha$ gives that with probability at least $1 - \delta$,

$$(\mathcal{L}_T(\eta^\alpha) - L_T(\eta^\alpha)) - (\mathcal{L}_T(\eta^*) - L_T(\eta^*)) \leq \sqrt{2T \log\left(\frac{10|\mathcal{H}^\alpha|T^2}{3\delta}\right)}$$

$$\implies (\mathcal{L}_T(\eta) - L_T(\eta)) - (\mathcal{L}_T(\eta^*) - L_T(\eta^*)) \leq \sqrt{2T \log\left(\frac{10|\mathcal{H}^\alpha|T^2}{3\delta}\right)}$$

$$+ \underbrace{(\mathcal{L}_T(\eta) - L_T(\eta)) - (\mathcal{L}_T(\eta^\alpha) - L_T(\eta^\alpha))}_{\text{discretization error}}.$$

The discretization error can be expanded and bounded as

$$\sum_{t=0}^{T-1} \left[ \mathbb{E}_{O \sim f_{\eta^*}(A_t)} \left[ \ell(A_t, O, \eta) - \ell(A_t, O, \eta^\alpha) \right] - \ell(A_t, O_t, \eta) + \ell(A_t, O_t, \eta^\alpha) \right] \leq 2\alpha T.$$

Since $\eta^*$ is a minimizer of $\mathcal{L}_T(\cdot)$, we have that with probability at least $1 - \delta$,

$$L_T(\eta^*) - L_T(\eta) \leq \sqrt{2T \log\left(\frac{10|\mathcal{H}^\alpha|T^2}{3\delta}\right)} + 2\alpha T.$$

We take the infimum over the size of $\alpha$-covers, which results in the bound

$$L_T(\eta^*) - L_T(\eta) \leq \sqrt{2T \log\left(\frac{10N(\mathcal{H}, \alpha, d_\mathcal{H})T^2}{3\delta}\right)} + 2\alpha T.$$

$\square$

Taking $\delta = \frac{1}{T}$ and plugging $\beta_T = \beta_T^*(\mathcal{H}, \delta, \epsilon_T^\mathcal{H})$ into the abstract regret bound in Theorem 2 proves the following main theorem.

**Theorem** (Expanded statement of Theorem 1)**.** *For all $T \in \mathbb{N}$,*

$$\text{Regret}(T) \leq 1 + \frac{1}{T} + \min\{\dim_{TE}(\mathcal{H}, \ell, \epsilon_T^\mathcal{H}), T\}$$

$$+ 2\sqrt{3\sqrt{2} \log\left(\frac{10N(\mathcal{H}, \alpha, d_\mathcal{H})T^2}{3\delta}\right)^{1/2} \dim_{TE}(\mathcal{H}, \ell, \epsilon_T^\mathcal{H})T^{3/2} + 6\dim_{TE}(\mathcal{H}, \ell, \epsilon_T^\mathcal{H})}.$$

*Proof.*

$$\text{Regret}(T)$$

$$\leq 1 + \frac{1}{T} + \min\{\dim_{TE}(\mathcal{H}, \ell, \epsilon_T^\mathcal{H}), T\} + 2\sqrt{3\dim_{TE}(\mathcal{H}, \ell, \epsilon_T^\mathcal{H})\beta_T^*(\mathcal{H}, \delta, \epsilon_T^\mathcal{H})T}$$

$$= 1 + \frac{1}{T} + \min\{\dim_{TE}(\mathcal{H}, \ell, \epsilon_T^\mathcal{H}), T\}+$$

$$+ 2\sqrt{3\dim_{TE}(\mathcal{H}, \ell, \epsilon_T^\mathcal{H})\left(\sqrt{2T \log\left(\frac{10N(\mathcal{H}, \epsilon_T^\mathcal{H}, d_\mathcal{H})T^2}{3\delta}\right)} + 2\epsilon_T^\mathcal{H}T\right)T}$$

$$= 1 + \frac{1}{T} + \min\{\dim_{TE}(\mathcal{H}, \ell, \epsilon_T^\mathcal{H}), T\}+$$

$$+ 2\sqrt{3\sqrt{2} \log\left(\frac{10N(\mathcal{H}, \alpha, d_\mathcal{H})T^2}{3\delta}\right)^{1/2} \dim_{TE}(\mathcal{H}, \ell, \epsilon_T^\mathcal{H})T^{3/2} + 6\epsilon_T^\mathcal{H} \dim_{TE}(\mathcal{H}, \ell, \epsilon_T^\mathcal{H})T^2}$$

$$\leq 1 + \frac{1}{T} + \min\{\dim_{TE}(\mathcal{H}, \ell, \epsilon_T^\mathcal{H}), T\}+$$

$$+ 2\sqrt{3\sqrt{2}\log\left(\frac{10N(\mathcal{H}, \alpha, d_{\mathcal{H}})T^2}{3\delta}\right)^{1/2}\dim_{TE}(\mathcal{H}, \ell, \epsilon_T^{\mathcal{H}})T^{3/2} + 6\dim_{TE}(\mathcal{H}, \ell, \epsilon_T^{\mathcal{H}})},$$

where the last inequality follows since $\epsilon_T^{\mathcal{H}} \leq 1/T^2$ by definition. □

The leading term in the regret bound is of order

$$T^{3/4}\left(\log N(\mathcal{H}, \epsilon_T^{\mathcal{H}}, d_{\mathcal{H}})\right)^{1/4}\sqrt{\dim_{TE}(\mathcal{H}, \ell, \epsilon_T^{\mathcal{H}})}.$$

**Remark 3.** As noted earlier on, while the order $\widetilde{O}(T^{3/4})$ on the time horizon $T$ may appear suboptimal compared to classical $\widetilde{O}(\sqrt{T})$ optimal rates for bandit learning with direct reward feedback, this slower rate is in fact a principled consequence of our minimal assumptions. Specifically, our analysis makes no structural assumptions on the hypothesis verifier loss $\ell$ beyond boundedness. If we have more structural knowledge of $\ell$, say, that it is $\alpha$-strongly convex, then the bound can be tightened to match the optimal order $\widetilde{O}(\sqrt{T})$. A notable instance is when $\ell$ is a squared loss. A refined analysis on the drift of conditional mean losses allows us to choose the confidence parameters $\beta_T$ for the version spaces to be of order $\widetilde{O}(\log(1/\delta))$, which results in the tight $\widetilde{O}(\sqrt{T})$ regret rate.

### B.4. Rate-Optimal Bound for Squared Loss

In this section, we consider a special case of a hypothesis verifier $\ell$, taking the discriminative example introduced in Section 3.3 and detailed in Section C.3.

**Theorem 3.** *Suppose $r_\eta(a) = \mathbb{E}_{o \sim f_\eta(a)}[g(a, o)]$ for some known $g : \mathcal{A} \times \mathcal{O} \to [0, 1]$ and $\ell(a, o, \eta) = (g(a, o) - r_\eta(a))^2 = (g(a, o) - \mathbb{E}_{o' \sim f_\eta(a)}[g(a, o')])^2$. Suppose for all $t \in \mathbb{N}$, $g(A_t, O_t) - \mathbb{E}_{O' \sim f_\eta(A_t)}[g(A_t, O')]$ conditioned on $(\mathcal{G}_t, A_t)$ is $\sigma$-sub-Gaussian. For all $T \in \mathbb{N}$, the regret of LLF-UCB satisfies*

$$\text{Regret}(T) \leq \widetilde{O}\left(\sqrt{T \log N(\mathcal{H}, \epsilon_T^{\mathcal{H}}, d_{\mathcal{H}}) \dim_{TE}(\mathcal{H}, \ell, \epsilon_T^{\mathcal{H}})}\right),$$

*where $N(\mathcal{H}, \epsilon_T^{\mathcal{H}}, d_{\mathcal{H}})$ denotes the $\epsilon_t^{\mathcal{H}}$-covering number of $\mathcal{H}$ based on the pseudo-metric $d_{\mathcal{H}}$, $\dim_{TE}(\mathcal{H}, \ell, \epsilon_T^{\mathcal{H}})$ denotes the $\epsilon_T^{\mathcal{H}}$-transfer eluder dimension of $\mathcal{H}$, and $\epsilon_T^{\mathcal{H}} = \max\left\{\frac{1}{T^2}, \min_{a \in \mathcal{A}} \inf\{|r_\eta(a) - r^*(a)| : \eta \in \mathcal{H}, \eta \neq \eta^*\}\right\}$.*

## C. Proofs for Supporting Lemmas and Propositions

### C.1. Proof for Proposition 1

*Proof.* Let $\tilde{\ell} = C_F \ell$. Let $d_{TE} = \dim_{TE}(\mathcal{H}, \tilde{\ell}, \epsilon)$ be the shorthand for the $\epsilon$-transfer eluder dimension of $\mathcal{H}$ with respect to $\tilde{\ell}$. Then, there exists a length $d_{TE}$ sequence of elements in $\mathcal{A}$ such that for some $\tilde{\epsilon} \geq \epsilon$, every action element is $\tilde{\epsilon}$-transfer independent of its predecessors. We denote such a sequence as $(a_0, \ldots, a_{d_{TE}-1})$. By definition of the transfer eluder dimension, for any $k \in \{0, \ldots, d_{TE} - 2\}$, there exists a pair of hypotheses $\eta, \eta' \in \mathcal{H}$ satisfying

$$\sum_{i=0}^{k}\left(\mathbb{E}_{o \sim f_{\eta'}(a_i)}[\tilde{\ell}(a_i, o, \eta)] - \tilde{\ell}_{\eta'}^{\min}(a_i)\right) \leq \tilde{\epsilon}^2$$

but $|r_\eta(a_{k+1}) - r_{\eta'}(a_{k+1})| > \tilde{\epsilon}$. Using the definition for reward-discriminative verifiers,

$$\sum_{i=0}^{k}(r_\eta(a_i) - r_{\eta'}(a_i))^2 \leq C_F \sum_{i=0}^{k}\left(\mathbb{E}_{o \sim f_{\eta'}(a_i)}[\ell(a_i, o, \eta)] - \ell_{\eta'}^{\min}(a_i)\right)$$

$$= \sum_{i=0}^{k}\left(\mathbb{E}_{o \sim f_{\eta'}(a_i)}[\tilde{\ell}(a_i, o, \eta)] - \tilde{\ell}_{\eta'}^{\min}(a_i)\right) \leq \tilde{\epsilon}^2.$$

By the definition of the (regular) eluder dimension, every action in the sequence $(a_0, \ldots, a_{d_{TE}-1})$ is $\epsilon$-independent of its predecessors. Therefore, $d_{TE} \leq \dim_E(\mathcal{R}, \epsilon)$ since the latter is the length of the longest sequence of independent actions. We may conclude that $\dim_E(\mathcal{R}, \epsilon) \geq \dim_{TE}(\mathcal{H}, C_F\ell, \epsilon)$.

□

### C.2. Proof for Lemma 5

**Lemma 5.** *Consider some $\bar{\mathcal{H}}$. Suppose $\min_{\pi \in \Pi} \max_{\eta \in \bar{\mathcal{H}}} r_\eta(\pi_\eta) - r_\eta(\pi) = 0$. Let $\hat{\pi}$ be a minimizer. Let $\mathcal{A}_\eta^*$ denote the set of optimal actions with respect to $r_\eta$. Then $\text{supp}(\hat{\pi}) \subseteq \mathcal{A}_\eta^*$, for all $\eta \in \bar{\mathcal{H}}$.*

*Proof.* We prove by contradiction. Suppose $\hat{\pi}$ takes some action $a'$ outside of $\mathcal{A}_\eta^*$ for some $\eta \in \bar{\mathcal{H}}$ with probability $p'$. Let $\pi' = \hat{\pi} - p'\mathbb{1}[a = a'] + p'\text{Unif}[a \in \mathcal{A}_\eta^*]$. Then it follows $r_\eta(\pi') > r_\eta(\hat{\pi})$, which is a contradiction. Therefore, $\text{supp}(\hat{\pi}) \subseteq \mathcal{A}_\eta^*$, for all $\eta \in \mathcal{H}$. $\qquad\square$

### C.3. Proof of the Discriminative Feedback Example

Suppose $r_\eta(a) = \mathbb{E}_{o \sim f_\eta(a)}[g(a, o)]$ for some known $g : \mathcal{A} \times \mathcal{O} \to [0, 1]$. Note that the reward mapping $\eta \mapsto r_\eta$ is known, but the reward function itself is still hidden from the agent (since $\eta^*$ is unknown). We define $\ell(a, o, \eta) := (g(a, o) - r_\eta(a))^2 = (g(a, o) - \mathbb{E}_{o' \sim f_\eta(a)}[g(a, o')])^2$, which gives

$$\mathbb{E}_{o \sim f_\eta(a)}[\ell(a, o, \eta')] = \mathbb{E}_{o \sim f_\eta(a)}\left[(g(a, o) - \mathbb{E}_{o' \sim f_{\eta'}(a)}[g(a, o')])^2\right].$$

One can easily verify that $\eta \in \arg\min_{\eta' \in \mathcal{H}} \mathbb{E}_{o \sim f_\eta(a)}[\ell(a, o, \eta')]$. With this definition, we have that

$$\begin{aligned}
|r_\eta(a) - r_{\eta'}(a)|^2 &= (\mathbb{E}_{o \sim f_\eta(a)}[g(a, o)] - \mathbb{E}_{o \sim f_{\eta'}(a)}[g(a, o)])^2 \\
&= (\mathbb{E}_{o \sim f_\eta(a)}[g(a, o) - \mathbb{E}_{o' \sim f_{\eta'}(a)}[g(a, o')]])^2 \\
&\leq \mathbb{E}_{o \sim f_\eta(a)}[(g(a, o) - \mathbb{E}_{o' \sim f_{\eta'}(a)}[g(a, o')])^2] \\
&= \mathbb{E}_{o \sim f_\eta(a)}[\ell(a, o, \eta')]
\end{aligned}$$

This shows the feedback is discriminative.

### C.4. Proof of Reasoning Example

**binary indicator of whether all steps are correct**     This problem is equivalent to a bandit problem with $|\mathcal{S}|^L$ arms. Here $f_\eta(a) = r(a)$, so the transfer eluder dimension reduces to the standard eluder dimension, which is bounded by the size of the action space.

**index of the first incorrect step**     Here we prove for $\epsilon < 1/2L$. Given the rubric of $\eta^*$, partition the action space into $L$ sets, where $\mathcal{A}_l = \{(s_1, \ldots, s_L)|s_1, \ldots, s_{l-1} \text{ are correct and } s_l \text{ is incorrect}\}$ for $l = 1, \ldots, L$, where $\mathcal{A}_0$ denotes sequences where $s_1$ is incorrect. By this definition, we have $\mathcal{A}_i \bigcap \mathcal{A}_j = \emptyset$, for $i \neq j$, and $\mathcal{A}^* \bigcup (\bigcup_{l=1}^L \mathcal{A}_l) = \mathcal{A}$, where $\mathcal{A}^* = \{a^*\}$

Suppose we have an independent action sequence $(a_1, \ldots, a_K)$ in the sense of Definition 3 where each action is $\epsilon$-independent of their predecessors. We show it can have no more than $|\mathcal{S}|$ actions from each $\mathcal{A}_l$ for $l \in [1, L]$. By definition of the feedback, for $a \in \mathcal{A}_l$, $f_\eta^*(a) = l$. Suppose we have more than $|\mathcal{S}|$ actions from $\mathcal{A}_l$. It implies that a token must be used twice at the $l$th position. Say it's $s_l$ and it's shared by $a^1, a^2 \in \mathcal{A}_l$. Then we show $a^2$ is $\epsilon$-dependent on $a^1$ when $\epsilon < 1/L$. For $\eta \in \mathcal{H}$, satisfying $\mathbb{E}_{o \sim f^*(a^0)}[|o - f_\eta(a^0)|^2/L^2] = |l - f_\eta(a^0)|^2/L^2 \leq \epsilon^2$, we have $l - L\epsilon \leq f_\eta(a^0) \leq l + L\epsilon$. Since $\epsilon < 1/2L$ and $f_\eta(a^0)$ is an integer, this implies $f_\eta(a^0) = l$. That is, for such an $\eta$ satisfying the constraint given by $a^0$, $s_l$ is incorrect. This implies $f_\eta(a^1) \leq l$. Therefore, $r_\eta(a^0) = r_\eta(a^1) = 0$.

Therefore, the length of independent action sequences is bounded by $|\mathcal{S}|L + |\mathcal{A}^*| = |\mathcal{S}|L + 1$.

**give correction for the first mistake**     In this case, the feedback not only returns the index of the first incorrect step $l$, but also reveals the correct reasoning action $s_l^*$. Let $a_\eta^* = (s_1(\eta), \ldots, s_L(\eta))$ denote the $L$ reasoning steps based on the hypothesis $\eta$. The reward function of any action $a$ and hypothesis $\eta$ is $r_\eta(a) = \mathbb{I}\{a_\eta^* = a\}$. For an action $a = (s_1, \ldots, s_L)$ and feedback $o := (l, s_l(\eta))$ generated based on $f_\eta(a)$, we have $s_j = s_j(\eta)$ for all $j < l$ and $s_l \neq s_l(\eta)$. Now, given any feedback $o := (l, s_l^*)$, we define the following loss $\ell(a, o, \eta) = \frac{1}{L}\left(\sum_{j=1}^{l-1} \mathbb{I}\{s_j(\eta) = s_j\} + \mathbb{I}\{s_l(\eta) = s_l^*\}\right)$. This verifer loss evaluates whether $\eta$ and $\eta'$ have the same first $l$ reasoning steps.

For $\epsilon < 1$, suppose an action sequence $(a_1, \ldots, a_K)$ where each action is $\epsilon$-independent of their predecessors. If action $a$ is $\epsilon$-independent, there exists $\eta, \eta'$ such that $\sum_{i=1}^K \mathbb{E}_{o_i \sim f_{\eta'}(a)}[l(a_i, o_i, \eta)] \leq \epsilon$ and $|r_\eta(a) - r_{\eta'}(a)| > \epsilon$. By definition of

the feedback and loss, we know $\eta,\eta'$ have the same initial $\max_i l_i$ reasoning steps. However, we know that $r_\eta(a) \neq r_{\eta'}(a)$ indicating at least one index $l > \max_i l_i$ where $s_l \in \{s_l(\eta), s_l(\eta')\}$ and $s_l(\eta) \neq s_l(\eta')$, resulting in feedback $o = (l, s_l(\eta'))$ for $a$. Thus, the sequence of indices in feedback $o_1, o_2, \ldots$ is monotonic. As we have $L$ reasoning steps, for any pair $\eta, \eta'$, the sequence length is bounded by $L$.

**demonstration** Here, the feedback directly demonstrates correct reasoning sequence $a^* = (s_1^*, \ldots, s_L^*)$ and is independent of the agent's action sequence. For action $a = (s_1, \ldots, s_L)$ and hypothesis $\eta$, we define the loss as $\ell(a, o, \eta) = \mathbb{I}\{o = a_\eta^*\}$. Therefore, for any $\eta, \eta'$ and $\epsilon < 1$, if $a$ satisfies: $\mathbb{E}_{o \sim f_{\eta'}(a)} \ell(a, o, \eta) \leq \epsilon$, we have $a_\eta^* = a_{\eta'}^*$, implying $r_\eta(a) = r_{\eta'}(a)$ for all $a \in |\mathcal{S}|^L$ and a transfer Eluder dimension of 1.

### C.5. Proof for Lemma 6

**Lemma 6.** *Suppose for some $t_0 \geq 0$, we have that $\min_{\pi \in \Pi} \max_{\eta \in \mathcal{H}_{t_0}} |r_\eta(\pi_\eta) - r_\eta(\pi)| = 0$ in Algorithm 1. Then for all $t > t_0$, $\min_{\pi \in \Pi} \max_{\eta \in \mathcal{H}_t} |r_\eta(\pi_\eta) - r_\eta(\pi)| = 0$.*

*Proof.* We prove by induction. Suppose the conclusion holds for $t > t_0$, we prove that it holds for $t + 1$ as well. At time $t$, the induction hypothesis implies that $\min_{\pi \in \Pi} \max_{\eta \in \mathcal{H}_t} |r_\eta(\pi_\eta) - r_\eta(\pi)| = 0$. Since $\mathcal{H}_{t+1} \subseteq \mathcal{H}_t$, $\max_{\eta \in \mathcal{H}_{t+1}} |r_\eta(\pi_\eta) - r_\eta(\pi)| \leq \max_{\eta \in \mathcal{H}_t} |r_\eta(\pi_\eta) - r_\eta(\pi)|$ for all $\pi \in \Pi$. Thus, $\min_{\pi \in \Pi} \max_{\eta \in \mathcal{H}_{t+1}} |r_\eta(\pi_\eta) - r_\eta(\pi)| \leq \min_{\pi \in \Pi} \max_{\eta \in \mathcal{H}_t} |r_\eta(\pi_\eta) - r_\eta(\pi)| = 0$. $\qquad\square$

## D. Additional Examples

We give an additional example of the hypothesis verifier below. In this example, reward-only learning requires exponential time ($2^L$), whereas the transfer eluder dimension is 1: LLF gives an exponential speedup.

**Example 2** (Bitwise feedback on 0-1 string)**.** Consider an action set $\mathcal{A} = \{0, 1\}^L$. The space of hypotheses $\mathcal{H}$ contains all possible length-$L$ 0-1 strings. Each hypothesis $\eta$ contains a particular fixed target string $s(\eta)$ and the corresponding text instruction to provide reward and feedback about the target. The reward function $r_\eta$ corresponding to a hypothesis $\eta$ is such that $r(a) = 1$ if $a = s(\eta)$ and $r(a) = 0$ otherwise. In other words, rewards are sparse and every suboptimal arm incurs a regret of 1. Feedback to an action $a = (a_1, \ldots, a_L)$ is bitwise, which tells in words the correctness of each bit in the 0-1 string (i.e. whether $a_i = s_i$ for $s(\eta) = (s_1, \ldots, s_L)$. Equivalently, we can abstract the feedback as $f_\eta(a) = (\mathbb{1}\{a_i = s_i\})_{i=1}^L$ and define the loss function $\ell(a, o, \eta) = \frac{1}{L} \sum_{i=1}^L \mathbb{1}\{o_i \neq \mathbb{1}\{a_i = s_i\}\}$ to measure the discrepancy between the feedback and the correctness indicated by hypothesis $\eta$. For any $\epsilon < \frac{1}{L}$, the transfer eluder dimension $\dim_{TE}(\mathcal{H}, \ell, \epsilon) = 1$, as for any action $a'$, the expected loss $\mathbb{E}_{O \sim f_{\eta'}(a')}[\ell(a', O, \eta)] < \frac{1}{L}$ iff $\eta = \eta'$.

## E. Extensions

### E.1. Special Case of Reward-Agnostic Feedback

Text feedback may contain information beyond what is relevant to the reward. In particular, one could imagine a special case, where feedback does not reveal much about the reward, but still provides enough to identify an optimal action over time. One simple example is when the feedback directly reveals the optimal action, regardless of the action chosen. In this case, the transfer eluder dimension as defined could be arbitrarily large, but ideally an efficient LLF agent should choose the optimal action in the following steps instead of trying to identify the mean reward for each action.

### E.2. Extension to Contextual Bandits

Our formulation can be modified slightly to accommodate learning with a context. While the feedback in the context-less setting may be viewed similar to a context, the main difference is that the optimal actions in the context-less setting do not change between iterations; on the other hand, in the contextual setting, the optimal actions in each time step depend on the context presented to the agent at that point. In a contextual problem, a Markov process $X_t$ independently takes values in a set $\mathcal{X}$ that the agent views as contexts. We may define the full set of actions to be the set of context-action pairs $\mathcal{A} := \{(x, a) : x \in \mathcal{X}, a \in \mathcal{A}(x)\}$, where $\mathcal{A}(x)$ is the set of available actions under the context $x$. Instead of having a fixed action space $\mathcal{A}$ across time, consider time-varying action sets $\mathcal{A}_t := \{(X_t, a) : a \in \mathcal{A}(X_t)\}$. At each time $t$, an action $a_t \in \mathcal{A}_t$ will be selected. In accordance, the policy $\pi = \{\pi_t | t \in \mathbb{N}\}$ is now a sequence of functions indexed by time, each

mapping the history $H_t = (\mathcal{A}_0, A_0, R_0, \ldots, \mathcal{A}_{t-1}, A_{t-1}, R_{t-1}, \mathcal{A}_t)$ to a distribution over $\mathcal{A}$ with support $\mathcal{A}_t$. Our analysis for the context-free setting directly carries over.

### E.3. Alternative Formulation of Feedback Generation

The LLF formulation we have presented so far assumes that feedback arises from a fixed mapping $\eta \mapsto f_\eta$ with each hypothesis $\eta \in \mathcal{H}$. While this "model-based" view simplifies both the design of exploration strategies and the complexity analysis via the transfer eluder dimension, it imposes a structural constraint that may be too restrictive in settings where feedback is generated by a more complex or even adversarial process. An alternative, entirely "model-free" formulation allows feedback to be generated arbitrarily from an oracle in a streaming fashion, without the need to explicitly model a feedback mapping $\eta \mapsto f_\eta$. Concretely, at each time $t$, the agent executes an action $A_t \in \mathcal{A}$ and observes feedback $O_t \in \mathcal{O}$. We denote the history of interactions as $I_t = (A_0, O_0, \ldots, A_t, O_t)$ and write $\mathcal{I}$ for the set of all possible histories. A (history-dependent) policy $\pi : \mathcal{I} \to \Delta(\mathcal{A})$ maps each history $h \in \mathcal{I}$ to a distribution over actions.

This streaming-oracle perspective subsumes both stochastic and adversarial feedback models, and can capture scenarios where the dependence on $\eta$ is unknown or too complex to parameterize. In this setting, one must replace the hypothesis-indexed complexity measures by complexity metrics defined directly over the space of oracles or possible histories. Although this general approach will likely incur additional technical overhead, it also broadens the applicability of our LLF framework to encompass richer feedback protocols beyond the hypothesis-testing paradigm. An interesting future direction is to develop performance guarantees under the more general feedback generation model.

### E.4. Relaxations of Assumption 3

Our verifier assumption can be naturally extended to allow noise and small approximation error in the verifier loss. In Assumption 3, unbiasedness is defined for the average loss over the distribution of observed feedback. In particular, an LLM-based hypothesis verifier may occasionally misjudge individual feedback instances, but as long as its errors are not systematically biased—i.e., it is correct on average over the distribution of feedback—it satisfies the assumption. If we assume a $\Delta$-approximately correct verifier to model potential LLM mistakes (formalized by modifying Assumption 3 to $\mathbb{E}_{O \sim f_\eta(a)}[\ell(a, O, \eta)] \leq \ell_\eta^{\min}(a) + \Delta$), then the regret bound acquires an additional linear bias term of order $O(\sqrt{\Delta})$. In other words, $\Delta$ controls how much the regret guarantee degrades under verifier error.

The regret analysis can also be extended to accommodate an approximate reward mapping $\tilde{r}_\eta$ with $\sup_{a,\eta} |\tilde{r}_\eta(a) - r_\eta(a)| \leq \delta$. In that case, the regret decomposition will accrue an additional approximation error $2\delta$ at each step, accumulating to an additive order $O(\delta T)$ term on the regret upper bound. Another subtlety this creates is the exploitation step, as an approximate minimax criterion no longer guarantees zero regret every step onwards. If we modify HELiX to take the exploitation step when the approximate minimax value is zero, i.e., $r_{\eta_p}(\pi_{\eta_p}) - r_{\eta_p}(\pi_p) = 0$, then the regret bound accrues an additional additive term $2\delta T$. Putting these together, the regret guarantee does degrade gracefully via an additive $O(\delta T)$ term. Concretely, suppose the algorithm observes, instead of $\eta \mapsto r_\eta$, a perturbed version $\eta \mapsto \tilde{r}_\eta$ that satisfies $\sup_{a \in \mathcal{A}, \eta \in \mathcal{H}} |r_\eta(a) - \tilde{r}_\eta(a)| \leq \delta$ uniformly for some $\delta > 0$. Define $\widetilde{U}_t(a) = \sup_{\eta \in \mathcal{V}_t} \tilde{r}_\eta(a)$, then $\sup_{a \in \mathcal{A}} |\widetilde{U}_t(a) - U_t(a)| \leq \delta$. Since the algorithm selects an action $A_t$ that maximizes $\tilde{U}_t(a)$, we have

$$
\begin{aligned}
r^*(\eta^*) - r^*(A_t) &\leq (U_t(a^*) - r^*(A_t)) \cdot \mathbb{1}\{\eta^* \in \mathcal{V}_t\} + \mathbb{1}\{\eta^* \notin \mathcal{V}_t\} \\
&\leq w_{\mathcal{V}_t}(A_t) \cdot \mathbb{1}\{\eta^* \in \mathcal{V}_t\} + \mathbb{1}\{\eta^* \notin \mathcal{V}_t\} + [U_t(a^*) - U_t(A_t)] \cdot \mathbb{1}\{\eta^* \in \mathcal{V}_t\} \\
&\leq w_{\mathcal{V}_t}(A_t) \cdot \mathbb{1}\{\eta^* \in \mathcal{V}_t\} + \mathbb{1}\{\eta^* \notin \mathcal{V}_t\} + [\widetilde{U}_t(a^*) - \widetilde{U}_t(A_t) + 2\delta] \cdot \mathbb{1}\{\eta^* \in \mathcal{V}_t\} \\
&\leq w_{\mathcal{V}_t}(A_t) \cdot \mathbb{1}\{\eta^* \in \mathcal{V}_t\} + \mathbb{1}\{\eta^* \notin \mathcal{V}_t\} + 2\delta \cdot \mathbb{1}\{\eta^* \in \mathcal{V}_t\}.
\end{aligned}
$$

Taking the expectation and summing over $t = 0, \ldots, T-1$, we obtain

$$
\mathrm{Regret}(T, \eta^*) \leq \sum_{t=0}^{T-1} \mathbb{E}\left[ w_{\mathcal{V}_t}(A_t) \cdot \mathbb{1}\{\eta^* \in \mathcal{V}_t\} + \mathbb{1}\{\eta^* \notin \mathcal{V}_t\} \right] + 2\delta T.
$$

It remains to handle the approximation in the exploitation step. If, in the exploitation step, $(\pi_p, \eta_p)$ satisfies

$$
\tilde{r}_{\eta_p}(\tilde{\pi}_{\eta_p}) - \tilde{r}_{\eta_p}(\pi_p) = 0,
$$

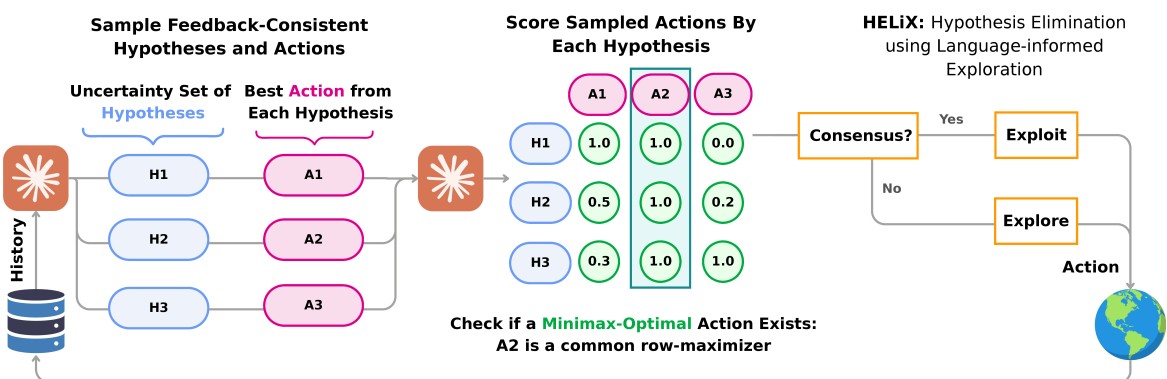

*Figure 4.* Our algorithm extends the traditional paradigm of model-based exploration to the LLM setting. Here, the "model" is represented by the LLM's thoughts, which we interpret as their hypotheses about the external world. We ground this think-then-act behavior in the interactive decision-making framework and introduce a new algorithm that conducts efficient exploration from language feedback.

where $\tilde{\pi}_\eta$ is an optimal policy under $\tilde{r}_\eta$, then by Lemma 5, $\tilde{r}_\eta(\pi_p) = \tilde{r}_\eta(\pi_\eta)$ for all $\eta \in \mathcal{H}_t$. Taking $\pi_p$ results in per-step regret at most

$$r_\eta(\pi_\eta) - r_\eta(\pi_p) \leq \tilde{r}_\eta(\pi_\eta) - \tilde{r}_\eta(\pi_p) + 2\delta \leq 2\delta$$

for all $\eta \in \mathcal{H}_t$. Thus, the regret incurred after taking the exploitation step is $O(\delta T)$.

## F. Experiment Details

### F.1. Environments

**BATTLESHIP** Battleship is a 2D grid environment where the agent must locate and sink three hidden ships within 20 turns. The agent fires at one cell per turn and receives hit/miss feedback, ship type (5-cell ship, 4-cell ship, and 3-cell ship), and a map showing all previous hits and misses. Success requires strategic exploration to find ships and exploitation to sink them efficiently. We use 20 scenarios (maps of ship layout) to evaluate all agents. We use a hidden per-step reward to evaluate an agent's performance. For instance, the feedback "a ship was hit but not sunk" corresponds to 0.5 point. We do not communicate this numerical reward information to the agent.

**MINESWEEPER** Minesweeper is a 2D grid puzzle with hidden mines. At each turn, the agent chooses to reveal one cell, aiming to uncover all safe cells within 20 turns without hitting a mine. Revealed cells show the number of adjacent mines, and a '0' triggers automatic revelations of surrounding safe cells. Sequential reasoning and updating of hypotheses based on observed clues are essential for success. Hidden rewards are calculated by assigning 0.2 to choosing a square that does not have a mine, and 1.0 to fully solving the game. Invalid moves incur a -0.2 penalty. The agent receives feedback in the form of a partially revealed map after each action.

### F.2. Baselines

**Chain-of-Thought (CoT)** LLMs with advanced reasoning capabilities can produce CoTs that often contain guesses and reasoning traces of the environment (Wei et al., 2022; DeepSeek Team, 2025; Gandhi et al., 2025). We propose to leverage LLMs' knowledge about the world to enhance decision-making. In particular, we treat an LLM's thinking tokens before deciding on an action as "hypotheses". These thinking tokens can be sampled by prompting the LLM to output its reasoning before an action with prompts in the form of "`<Hypothesis> <Action>`". Using a targeted prompt, we sample a single action in the style of hypothesis (thinking tokens) then act, similar to ReAct (Yao et al., 2023).

**CoT + epsilon-greedy** We implement an epsilon-greedy exploration baseline. We use the "propose exploration action prompt $(\pi_{ref})$" in Section F.8 to obtain a random action. We then use CoT as described above to get a greedy action. Because in both our battleship and minesweeper environment, we only take at most 20 interaction steps, we set the initial $\epsilon$ to be fairly large, and then anneal based on step count $\epsilon_t = \epsilon_{t-1} * \max(0, 1 - \frac{t}{20})$, where $\epsilon_0 = 0.5$.

## F.3. `HELiX` Implementation Details

**`HELiX`.** We provide a practical implementation of `HELiX` using LLMs. We provide the pseudocode for the practical implementation in Algorithm F.9 and illustrate a corresponding flow-graph in Figure 4. The algorithm takes as inputs the following LLM-based components:

1. $\pi_{\mathrm{LLM}} : \bigcup_{t=0}^{\infty}(\mathcal{A} \times \mathcal{O})^t \to \Delta(\mathcal{H} \times \mathcal{A})$. This is an LLM with a chain-of-thought prompt that asks it to analyze the current observation through thinking tokens and produce a valid action. We may view this policy as producing the best action conditioned on a hypothesis consistent with the feedback history.

2. $R_{\mathrm{LLM}} : \mathcal{H} \times \mathcal{A} \to [0, 1]$. This is a reward mapping to evaluate how good/bad the action is under a given hypothesis. We implement this by prompting an LLM to score an action conditioned on a sampled hypothesis. This can be viewed as a hypothesis-conditioned reward model.

`HELiX` (Algorithm 1) maintains a hypothesis space $\mathcal{H}_t$ at iteration $t$, which contains all hypotheses $\eta$ that are consistent with observed feedback. Then, `HELiX` searches over all possible policies by computing $\pi_p$ and $\pi_o$. We approximate these two steps with finite sets of candidates, $\hat{\mathcal{H}}_t$ and $\hat{\mathcal{A}}_t$, respectively.

We make the assumption that state-of-the-art LLMs are capable of producing valid hypotheses when instructed with a chain-of-thought prompt and history. In other words, they provide hypotheses that are plausible explanations of the interaction history of actions and feedback. At each step, we use $\pi_{\mathrm{LLM}}$ to produce hypothesis-conditioned actions. We first ask the LLM to generate a diverse set of hypotheses. For each hypothesis, we prompt an LLM to generate corresponding optimal actions. Unlike a common chain-of-thought approach that asks LLMs to produce only one hypothesis and one action, we ask the LLM to output $N$ hypotheses and actions. This set of hypotheses accounts for the agent's uncertainty about the environment. For computational efficiency, we sample these $N$ hypotheses and actions in one LLM call rather than $N$ calls, introducing conditional dependencies between them. These LLM calls produce an approximate hypothesis space $\hat{\mathcal{H}}_t$ of size $N$ and an approximate policy space $\hat{\mathcal{A}}_t$ (of deterministic actions) of size $N$.

We approximate the minimax and maximization steps in Algorithm 1 with $\hat{\mathcal{H}}_t$ and $\hat{\mathcal{A}}_t$. Concretely, we construct a score matrix $S_t \in [0, 1]^{N \times N}$ whose entries $[S_t]_{\eta, a}$ correspond to the reward of hypothesis-action pairs $(\eta, a)$. The rows of this score matrix correspond to hypotheses in $\hat{\mathcal{H}}_t$ and columns correspond to actions in $\hat{\mathcal{A}}_t$. This matrix is visualized in the middle portion in Figure 4. We use the reward mapping $R_{\mathrm{LLM}}$ to produce scores. The diagonal entries of $S_t$ are close to 1.0 because the action $a_i$ conditionally sampled from $\eta_i$ should be scored the highest under $\eta_i$. Given the score matrix $S_t$, we first check whether the exploitation step in Algorithm 1 can be triggered. We do so by first forming a set of actions that are optimal to all hypotheses:

$$\hat{\mathcal{A}}_t^* \leftarrow \bigcap_{\eta \in \hat{\mathcal{H}}_t} \arg\max [S_t]_\eta .$$

If a given action $a^\star$ satisfies $R_{\mathrm{LLM}}(\eta, a^\star) \geq R_{\mathrm{LLM}}(\eta, a)$ for all $\eta \in H_t$ and $a \in A_t$, then $a^*$ is identified as a consensus action and exploited immediately. This corresponds to the exploitation step in the theoretical Algorithm 1, and we visualize this step in Fig. 6, Panel A. By Lemma 5, if an action solves the minimax problem, it must also be an optimal action for all remaining hypotheses simultaneously. If there are multiple consensus actions, we perform tie-breaking detailed below. This step is implemented as a set intersection operation over the sets of highest-scoring actions from each hypothesis, with an illustrative example in Fig. 6, Panel B.

> **if** $\hat{\mathcal{A}}_t^* \neq \varnothing$ **then**
> $\quad A_{t+1} \leftarrow \texttt{tie-break}(\hat{\mathcal{A}}_t^*)$   // Exploitation step
> **else**
> $\quad A_{t+1} \leftarrow \underset{a \in \hat{\mathcal{A}}_t}{\arg\max} \left( \underset{\eta \in \hat{\mathcal{H}}_t}{\arg\max} [S_t]_\eta \right)$   // Exploration step
> **end if**

## F.4. Additional Results

We implement two variants of `HELiX` with slightly different action selection procedures.

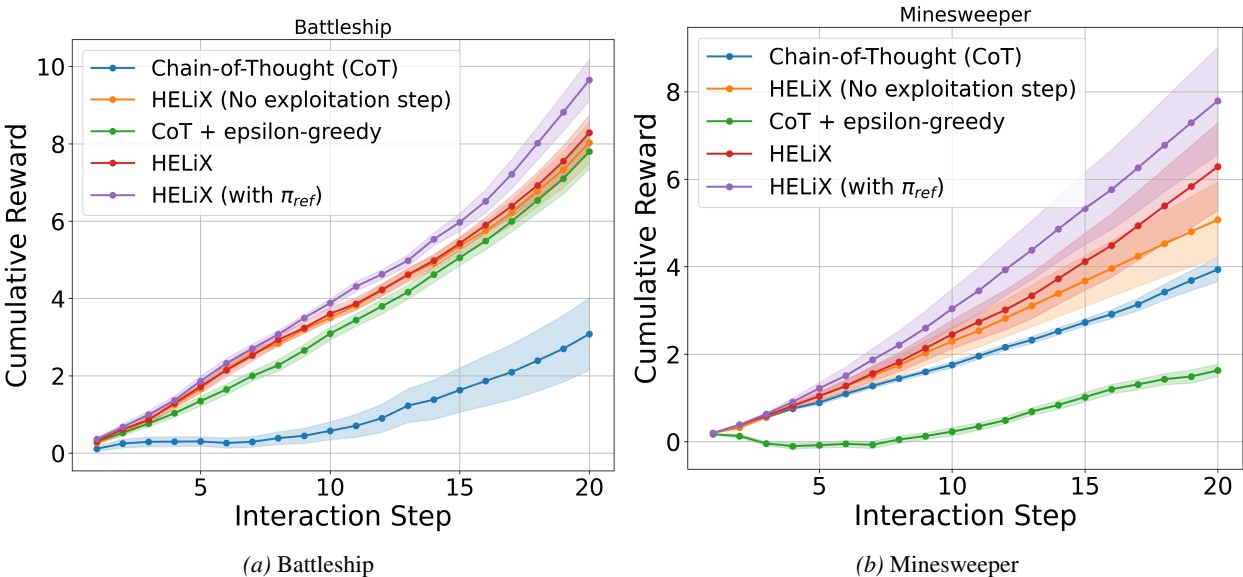

*(a)* Battleship          *(b)* Minesweeper

*Figure 5.* We show the cumulative reward that the agent is able to obtain during a fixed number of interactions with the environment. Shaded area represents the standard error of cumulative reward across different scenarios. We use Claude-Sonnet-3.5 v2.

**HELiX (No exploitation step).** We use the same procedure as HELiX to generate $N$ actions and $N$ hypotheses, followed by scoring each action under every hypothesis. Unlike in HELiX, we directly select actions with the highest score across all hypotheses. If there are multiple actions, we tie-break by preferring hypotheses and actions generated earlier in the output.

**HELiX (With $\pi_{\mathrm{ref}}$).** $\pi_{\mathrm{ref}} : \varnothing \to \Delta(\mathcal{A})$ is a user-provided reference policy to sample actions, analogous to a baseline policy. The design of the reference policy may vary. In this work, we adopt a random reference policy by asking an LLM to produce a set of random actions that are different from those generated by $\pi_{\mathrm{LLM}}$. These random actions are only used if there are ties during the exploration step. We apply a tie-breaking step by re-scoring with a reference policy. We sample $M$ actions from $\pi_{\mathrm{ref}}$: $\tilde{a} \sim \pi_{\mathrm{ref}}$ and use the same $R_{\mathrm{LLM}}$ to score each one conditioned on the sampled feedback-consistent hypotheses $\hat{\mathcal{H}}_t$, forming a rectangular score matrix $S_t \in [0,1]^{N \times (N+M)}$.

> **if** $\hat{\mathcal{A}}_t^* \neq \varnothing$ **then**
>     $A_{t+1} \leftarrow \texttt{tie-break}(\hat{\mathcal{A}}_t^*)$    // Exploitation step
> **else**
>     $\tilde{\mathcal{H}}_t \leftarrow \arg\max_{\eta \in \hat{\mathcal{H}}_t} \left( \max [S_t]_\eta \right)$
>     // Exploration step with re-scoring:
>     $A_{t+1} \leftarrow \arg\max_{a \in \hat{\mathcal{A}}_t} \left( \max_{\eta \in \tilde{\mathcal{H}}_t} \left[ [S_t]_{\eta,a} - \mathbb{E}_{\tilde{a} \sim \pi_{\mathrm{ref}}} \left[ [S_t]_{\eta,\tilde{a}} \right] \right] \right)$
> **end if**

The re-scoring or re-centering step is widely used in RL, such as baseline methods (Weaver and Tao, 2001; Sutton and Barto, 2018), ReMax (Li et al., 2024), RLOO (Ahmadian et al., 2024), and GRPO (DeepSeek Team, 2025). This procedure interprets the score as the advantage of an action $a$ relative to those sampled from a reference policy $\pi_{\mathrm{ref}}$, under a given hypothesis $\eta$. There are multiple reasons why an advantage is useful for tie-breaking: 1) LLMs may not score consistently across hypotheses. Comparing score differences can help cancel out these inconsistencies. 2) When we use a uniformly random $\pi_{\mathrm{ref}}$, the advantage implicitly examines the quality of the hypotheses and favors more discriminative ones. A permissive hypothesis that assigns approximately the same score to all actions (e.g., "Fire a shot anywhere on the map") lacks discriminative power. In contrast, a discriminative hypothesis assigns higher scores to actions that align with its intent (e.g., "Fire a shot along the edge of the map"), yielding a higher advantage over random actions. With re-scoring, we favor actions with high advantages over random actions. The re-scoring only changes the exploration step, where we use the $\pi_{\mathrm{ref}}$ to help decide which step to take.

We plot the cumulative reward of this additional variant along with those of previously introduced agents in Figure 5. In

## HELiX Explore-Exploit Decision Rule

Panel A and B represent how HELiX decides which action to take based on different score-matrix outcomes.

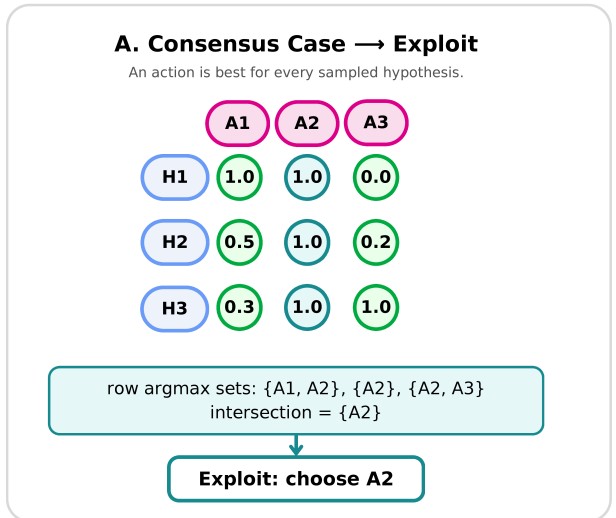
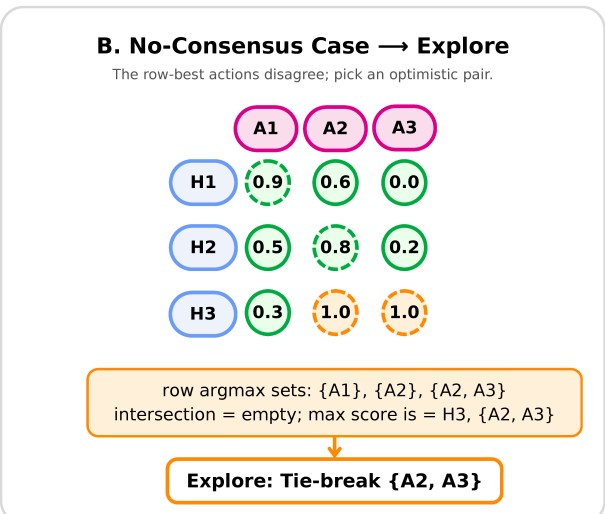

*Figure 6.* **HELiX Explore-Exploit Decision Rule.** The HELiX algorithm selects actions based on a score matrix over sampled hypotheses and candidate actions. Two panels represent two possible score-matrix outcomes. **(A) Consensus case (Exploit):** If the row-wise argmax sets share a common action across all hypotheses, that action is chosen. **(B) No-consensus case (Explore):** If the row-wise argmax sets have an empty intersection, the algorithm explores by selecting the action corresponding to the globally highest score. If multiple actions have the same highest score, we tie-break by preferring hypotheses and actions generated earlier in the output.

BATTLESHIP and MINESWEEPER, HELiX with $\pi_{\text{ref}}$ performs significantly better than the CoT baseline and other variants of HELiX, showing empirical benefits of the explicit re-scoring.

**A Note on Tie-breaking.** If ties remain after re-scoring (or without re-scoring, in the case where we don't use $\pi_{\text{ref}}$), we further tie-break by preferring hypotheses and actions generated earlier in the output. This is due to empirical observations that LLMs have a preference to produce the best plan and action first, followed by less likely plans and actions (Dracheva and Phillips, 2024).

### F.5. Additional Studies on Assumption Satisfaction

To give a sense of how Assumption 3 connects to our experiments, we evaluate a key implication of it: that the shrinking hypothesis set remains consistent with the true hypothesis along the trajectory. Concretely, if Assumption 3 approximately holds, then as the algorithm updates and prunes the hypothesis set, the true hypothesis should almost always remain included. To test this, we manually annotated 10 trajectories from the Battleship experiment. For each step in each trajectory, we checked whether the candidate set of hypotheses produced by HELiX contained the true hypothesis.

Table 3 reports (i) the average fraction of steps within a trajectory for which the true hypothesis is included ("Assumption 3 satisfaction (trajectory-averaged)"), and (ii) the fraction of all annotated steps across trajectories for which the true hypothesis is included ("Assumption 3 satisfaction (step-averaged)"). Both metrics are above 95%, empirically supporting that a key implication of Assumption 3 holds to a good approximation in our experiment.

*Table 3.* Empirical verification of Assumption 3 satisfaction rate in the Battleship experiment

| | |
|---|---|
| Assumption 3 satisfaction (trajectory-averaged) | 96.26% |
| Assumption 3 satisfaction (step-averaged) | 95.98% |

### F.6. Discussion on Computational Cost

HELiX incurs a computational cost on sampling that scales with the number of top hypotheses $K$ that we propose at each turn, and if we only look at the algorithm, building the score matrix through "approximate minimax game" does incur $K^2$ LLM calls. However, this is not as costly as it might seem due to several reasons:

**Better algorithms can lead to solving the game faster (i.e., early termination):** In all the games, we allow agents to act many more steps than necessary to solve the game. A better algorithm can often lead to quicker terminations. We verify this with the token count of our algorithm in Table 4.

**Parallelization of feedback-consistent hypotheses and actions sampling:** In practical implementations, we can sample hypotheses and verify consistency in parallel, reducing $O(K^2)$ calls to $O(K)$ in approximate minimax game and $O(K)$ to $O(1)$ when sampling feedback-consistent hypotheses and actions.

**Efficient use of tokens through prefix caching:** In practical implementations, we could leverage advanced inference techniques like prefix caching where instructions sharing the same prefix sequence can be stored and loaded as KV cache without re-computing. In HELiX, many LLM calls share common observations and judgments, significantly reducing the actual tokens needed. Since we mainly use the experiment to illustrate and instantiate one practical implementation, we did not implement prefix caching or parallel sampling. Table 4 mostly demonstrates that even without such advanced techniques, just by exploring the environment better, we still avoided exponential cost blowup (instead of 9x the tokens of the baseline, we are 3.73x to 4.02x the token count of baseline).

*Table 4.* Token count comparison for Battleship and Minesweeper

| Method | Token Count | Comparison |
|---|---|---|
| **Battleship** | | |
| Baseline | 698,873 | 1x |
| HELiX (no exploitation step) (explore optimistically) | 3,151,173 | 4.51x ($K = 3$ hypotheses) |
| HELiX | 2,812,856 | **4.02x** ($K = 3$ hypotheses) |
| **Minesweeper** | | |
| Baseline | 553,389 | 1x |
| HELiX (no exploitation step) (explore optimistically) | 2,538,838 | 4.59x ($K = 3$ hypotheses) |
| HELiX | 2,064,156 | **3.73x** ($K = 3$ hypotheses) |

### F.7. Reasoning Traces for Battleship

Here we show a concrete example on how the algorithm works and what the LLM generates.

---

**Transcript on Multi-Hypothesis Evaluation (Battleship)**

**Prompt:** Hit at D2! You have hit a ship, which occupies 4 cells in the grid. Here is how the board looks now:

```
    1   2   3   4   5   6
A   M   M   M   M   .   M
B   M   X   X   M   M   .
C   M   X   X   .   M   .
D   .   X   X   M   .   .
E   .   .   X   .   .   .
F   .   .   X   .   .   .
```

========================
Now you have a new task. You are given a hypothesis (thought/instruction) and actions. You need to evaluate if how good or bad the action is given the hypothesis.
Hypothesis:
 Alternatively, a 2-cell ship might be placed horizontally in the A5-B6 area, as this is one of the few remaining areas large enough to fit it. A5 would be worth checking.
Rate all the actions individually based on whether the action is aligned with the hypothesis.
Action 1: <action>E2</action>
Action 2: <action>E5</action>
Action 3: <action>A5</action>
Make sure the score you assign is between 0 and 1. Please provide your scores in the following format: ...

---

Response:

Action 1 for the Hypothesis:

<think> E2 is not aligned with the hypothesis at all. The hypothesis suggests checking A5 area for the 2-cell ship, but E2 is in a completely different part of the board. Additionally, we already have significant ship presence in the middle columns. </think><score> 0.1 </score>

Action 2 for the Hypothesis:

<think> E5 is somewhat in the direction of exploring new areas for remaining ships, but it's not in the specific A5-B6 area mentioned in the hypothesis. It's not directly supporting the hypothesis's suggested strategy. </think><score> 0.3 </score>

Action 3 for the Hypothesis:

<think> A5 is exactly what the hypothesis suggests checking. It's directly aligned with the reasoning that the 2-cell ship might be placed horizontally in the A5-B6 area, and it's one of the few remaining unexplored areas that could fit a 2-cell ship. </think><score> 1.0 </score>

---

**Transcript on Action Proposal (Battleship)**

Prompt: Hit at A5! You have hit a ship, which occupies 5 cells in the grid.
Here is how the board looks now:

```
    1   2   3   4   5
A   .   .   X   X   X
B   .   X   X   X   X
C   .   M   X   X   .
D   .   .   M   .   .
E   .   .   M   .   .
```

Given the information above, please propose 3 actions, make sure each of them is a valid action in the environment, and is reasonable.
Make sure your thoughts and hypotheses are diverse and unique. However, actions are allowed to be the same if they are the best possible action under that hypothesis.
Please provide your actions in the following format:
Action 1: <think> ...</think><answer> your first action</answer>
...
Action 3: <think> ...</think><answer> your 3rd action</answer>

Response:
Action 1: <think> Looking at the pattern of hits for the 5-cell ship, we can see it extends from A3 to A5. Since 5-cell ships are 5 cells long and we've found 3 cells, we should check A2 as it's likely part of the 5-cell ship.</think><answer> A2</answer>

Action 2: <think> If the 5-cell ship is not extending westward, it might extend to the east. Even though this is less likely since it would make the ship overlap with known hits, we should consider A1 to eliminate this possibility.</think><answer> A1</answer>

Action 3: <think> Since we've hit three cells of the ship (A3, A4, A5), and this ship is 5 cells long, if it's not extending horizontally, it must extend vertically. Therefore, checking C5 could reveal if the 5-cell ship extends downward from A5.</think><answer> C5</answer>

## F.8. Prompt Templates

---

**Propose Action Prompt ($\pi_{\text{LLM}}$)**

Given the information above, please propose some hypotheses and act according to those hypotheses.
You can propose at most {num_actions} hypotheses.
Please propose a reasonable number of hypotheses – each hypothesis represents what you think.
Please provide your actions in the following format:
Action 1: <think>...</think><answer>action 1</answer>
...
Action {num_actions}: <think>...</think><answer>your {num_actions}th action</answer>

---

**Propose Exploration Action Prompt ($\pi_{\text{ref}}$)**

Given the information above, please propose {num_actions} completely different and unexpected actions. These should be valid in the environment but should explore unusual or creative approaches.
Try to think outside the box and propose actions that might not be immediately obvious or conventional.
Here are the actions you have already proposed:
{actions}
Please avoid proposing the same actions.
Please provide your actions in the following format:
Action 1: <think>...</think><answer>your first random/exploratory action</answer>
...
Action {num_actions}: <think>...</think><answer>your {num_actions}th random/exploratory action</answer>

---

**Hypothesis-Conditioned Value Function Prompt ($V_{\text{LLM}}$)**

{task description}
========================
Now you have a new task. You are a given a hypothesis (thought/instruction) and actions. You need to evaluate how good or bad the action is given the hypothesis.

Hypothesis:
<think>
{hypothesis}
</think>

Rate all the actions indiviually based on whether the action is aligned with the hypothesis.

Action {action_idx}: <action>{action}</action>

Make sure the score you assign is between 0 and 1. Please provide your scores in the following format:

Action 1 for the Hypothesis:
<think>... </think>
<score>...</score>
...
Action {num_actions} for the Hypothesis:
<think>... </think>
<score>...</score>

---

## F.9. Code Implementation

We provide a high-level code snippet that demonstrates how we implement the algorithm below. We omit the implementation details of methods involving LLM calls.

**HELiX Python Code**

```python
import numpy as np

class HELiX:

    def select_action(self, observation, hypotheses, actions, random_actions):

        if random_actions is not None:
            actions = actions + random_actions  # evaluate on all actions

        # Create a matrix to store scores for each hypothesis-action pair
        score_matrix = np.zeros((len(hypotheses), len(actions)))

        # Fill the score matrix by evaluating each hypothesis-action pair
        for h_idx, hypothesis in enumerate(hypotheses):
            scores = self.evaluate_multi_hypotheses(observation, hypothesis, actions)
            score_matrix[h_idx] = scores

        # ======== Exploitation step: consensus check =======
        consensus_action = self.consensus_action(score_matrix, actions)
        if consensus_action is not None:
            return consensus_action

        # ====== UCB elimination ======
        score_matrix, hypotheses, actions = self.ucb_hypothesis_elimination(score_matrix
        .copy(), hypotheses, actions)

        # ====== (Re-scoring +) Exploration =====
        best_hypothesis, best_action, best_overall_score, best_action_indices = self.
        tie_breaking(score_matrix, hypotheses, actions)

        return best_action

    def consensus_action(self, score_matrix, actions):
        max_scores_per_row = np.max(score_matrix, axis=1)
        action_sets = []
        for i in range(score_matrix.shape[0]):
            action_sets.append(np.where(score_matrix[i] == max_scores_per_row[i])[0].
        tolist())
        # Convert each sublist to a set
        action_sets = [set(actions) for actions in action_sets]

        # Find the intersection of all sets
```

```python
40          overlapped_actions = reduce(lambda x, y: x.intersection(y), action_sets)

41

42          # Convert back to list if needed
43          overlapped_actions_list = list(overlapped_actions)

44

45          if len(overlapped_actions_list) == 0:
46              return None
47          else:
48              # randomly choose one
49              random_index = np.random.choice(len(overlapped_actions_list))
50              return actions[overlapped_actions_list[random_index]]

51

52      def tie_breaking(self, score_matrix, hypotheses, actions, random_actions=[]):

53

54          # ====== Optional Re-Scoring =========
55          # Calculate average scores only for random actions
56          num_regular_actions = len(actions) - len(random_actions)

57

58          # avg(random actions)
59          action_avg_scores = np.mean(score_matrix[:, num_regular_actions:], axis=1,
         keepdims=True)
60          normalized_score_matrix = score_matrix - action_avg_scores

61

62          # eliminate hypothesis again, to prevent ties
63          normalized_score_matrix, hypotheses, actions = self.ucb_hypothesis_elimination(
         normalized_score_matrix.copy(), hypotheses, actions)

64

65          best_hypothesis, best_action, best_overall_score, best_action_indices = self.
         two_tiered_argmax_sampling(
66              normalized_score_matrix, hypotheses, actions
67          )

68

69          return best_hypothesis, best_action, best_overall_score, best_action_indices

70

71      def ucb_hypothesis_elimination(self, score_matrix, hypotheses, actions):
72          # Get the maximum score for each row (hypothesis)
73          max_scores_per_row = np.max(score_matrix, axis=1)

74

75          # Find the highest score value
76          highest_score = np.max(max_scores_per_row)

77

78          # Get indices of rows that have the highest score
79          selected_row_indices = np.where(max_scores_per_row == highest_score)[0]

80

81          # Select the hypotheses corresponding to these rows
82          selected_hypotheses = [hypotheses[i] for i in selected_row_indices]
83          # we only eliminate hypotheses, not actions

84
```

```
85          # Create a new score matrix with only the selected rows and columns
86          new_score_matrix = score_matrix[selected_row_indices, :]
87
88          return new_score_matrix, selected_hypotheses, actions
89
90      def two_tiered_argmax_sampling(self, score_matrix, hypotheses, actions):
91          # we take the highest score hypothesis, then sample its highest action
92          assert score_matrix.shape[0] == len(hypotheses)
93
94          best_hypo_idx = np.argmax(np.max(score_matrix, axis=1))  # bias towards first
        hypothesis
95          best_action_idx = np.argmax(score_matrix[best_hypo_idx, :])
96
97          best_action = actions[best_action_idx]
98          best_hypothesis = hypotheses[best_hypo_idx]
99          best_overall_score = score_matrix[best_hypo_idx, best_action_idx]
100
101          return best_hypothesis, best_action, best_overall_score, [best_action_idx]
```

