# OpenReview forum: "Formalizing Learning from Language Feedback with Provable Guarantees"
_ICML.cc/2026/Conference — ICML 2026 regular_

### Official Review · Reviewer_7vGa · 2026-03-06

**Soundness:** 4
**Presentation:** 2
**Significance:** 3
**Originality:** 3
**Overall Recommendation:** 4
**Confidence:** 4

**Summary:**

This paper proposes a mathematical formulation of learning from language feedback. The authors take a hypothesis testing perspective, and introduce the concept of y hypothesis verifier to check the consistency of hypotheses with language feedback. Based on this, the transfer eluder dimension is introduced as a new learning complexity measure for learning from language feedback. The authors then propose a simple no-regret algorithm and conduct experiments in Wordle, Battleship and Minesweeper setups.

**Compliance With Llm Reviewing Policy:**

Affirmed.

**Final Justification:**

In the rebuttal, the authors clarified the scope, which is appreciated. The other concerns I had, namely, that assuming a hypothesis verifier is restrictive and that HeLiX is not using an explicit hypothesis verifier in the experiments persist.

I decided to keep my score that is weakly in favor of acceptance.

**Key Questions For Authors:**

1. Your formulation relies heavily on the hypothesis verifier to inrepret the feedback and eliminate inconsistent hypotheses. It is unclear to me how such a reliable verifier could be constructed in practice and validated.
2. Could you please clarify the scope of your formulation again here. From my understanding you focus on a bandit-style setting. Also, you do not consider model training. Could you briefly clarify the intended scope, please.
3. I'm curious to hear whether you yourself view your experiments (with the practical implementation of HELiX) as validating your theoretical findings, or are you evaluating a heuristic agent/algorithm design that is only inspired by your formal work in your experiments?

**Limitations:**

Mostly yes. Though, it would be helpful if the paper discussed its scope more explicitly.

**Strengths And Weaknesses:**

### Strengths
1. Formalizing learning from language feedback is an important and timely research problem. It is difficult as language feedback do not admit the clear structure and clean mathematical form of traditional feedback sources, which is why it is so hard to find good models and assumptions that are not too restrictive.
2. The transfer eluder dimension is a very intuitive and interesting complexity measure in my opinion. However, the usefulness of the concept depends heavily on the hypothesis verifier and the surrounding assumptions. This is where, in my opinion, some of the key weaknesses of the framework currently lie and I'm not yet fully convinced by it.

### Weaknesses
1. **Scoping, Positioning in the Literature**: At first reading, the scope of the work was unclear to me. The intrdouction reads as if the paper provides a formal model and guarantees for LLF *in general*. It only becomes clearer later (and after re-reading passages) that the focus is on an in-context learning and "bandit-style" setting (in contrast to iterative/multi-turn). For example, it also wasn't clear to me at the beginning that the paper does not consider model training with language feedback. The fact that Related Work is only discussed in Section 6 contributes to the lack of contextualization and scoping.
2. **Hypothesis Verifier**: Assumption 2 regarding the hypothesis verifier is restrictive, and it is very unclear whether it is a practical assumption. In the LLM implementation of HeLiX in Section 4.2, the hypothesis verifier is not even explicitly designed, but implicitly derived from prompting an LLM to sample thinking traces (as hypotheses) that are already consistent with observed feedback. This is so different from the theoretical formulation that I wonder about whether the theory actually informs practice here.
3. **Unbiased Feedback**: Assumption 3 is another strong assumption. Language feedback can be ambiguous, imperfect, noisy etc., and the question of its effect is one of the most interesting ones in LLF. By imposing Assumption 3 the paper avoids these challenges.

---

> ### Author Rebuttal · Authors · 2026-03-31
>
> Thank you for the positive review and constructive feedback.
>
> > At first reading, the scope of the work was unclear to me. The introduction reads as if the paper provides a formal model and guarantees for LLF in general. Could you briefly clarify the intended scope, please.
>
> Thank you for pointing out the source of misunderstanding. We will update the following sentence in the introduction: We address this gap by introducing a new, formal mathematical framework of Learning from Language Feedback (LLF) for single-turn in-context learning.
>
> > Assumption 2 regarding the hypothesis verifier is restrictive, and it is very unclear whether it is a practical assumption. Your formulation relies heavily on the hypothesis verifier to interpret the feedback and eliminate inconsistent hypotheses. It is unclear to me how such a reliable verifier could be constructed in practice and validated.
>
> As an implicit construction of the hypothesis verifier loss, we tuned prompts to ensure that hypothesis verification works robustly. Alternatively, we may construct an explicit hypothesis verifier loss via training or prompting an LLM, and use it to filter out inconsistent hypotheses. To validate our implicit construction, we perform a study in Appendix F.7, showing that Assumption 3 approximately holds in the Battleship experiment.
>
> > Assumption 3 is another strong assumption. Language feedback can be ambiguous, imperfect, noisy etc., and the question of its effect is one of the most interesting ones in LLF.
>
> Our verifier assumption can be naturally extended to allow noise and small approximation error in the verifier loss. In Assumption 3, unbiasedness is defined for the average loss over the distribution of observed feedback. In particular, an LLM-based hypothesis verifier may occasionally misjudge individual feedback instances, but as long as its errors are not systematically biased—i.e., it is correct on average over the distribution of feedback—it satisfies the assumption. If we assume a $\Delta$-approximately correct verifier to model potential LLM mistakes (formalized by modifying Assumption 3 to $E_{O\sim f_\eta(a)}[\ell(a,O,\eta)] \le \ell_\eta^{\min}(a) + \Delta$), then the regret bound acquires an additional linear bias term of order $O(\sqrt{\Delta})$. In other words, $\Delta$ controls how much the regret guarantee degrades under verifier error. We will add this discussion to the paper.
>
> > I'm curious to hear whether you view your experiments (with the practical implementation of HELiX) as validating your theoretical findings, or are you evaluating a heuristic agent/algorithm design that is only inspired by your formal work in your experiments?
>
> Both claims are to some extent valid. In this work, we formalize LLF problems, then design and analyze a principled algorithm derived from first principles and motivated by our theory. In theory, a greedy algorithm ought not work well for LLF.
>
> To validate this theoretical observation using experiments, we need to utilize LLMs and hence seek approximations of the theory version. The approximation nonetheless should be based on some heuristics, or as you put it, “inspired by the theoretical version”, but not in a loose sense. We tried our best to find the right approximation, as explained in Section 4.2.
>
> That said, we acknowledge that the goal of our experiments is mainly to validate ideas and inspire future work.

---

> > ### Author Rebuttal · Reviewer_7vGa · 2026-04-01
> >
> > Thank you for your response.
> >
> > I appreciate you clarifying the scope. The other main reservations I had remain, namely, assuming a hypothesis verifier is restrictive and that HeLiX is not using an explicit hypothesis verifier in your experiments but prompting an LLM to generate hypothesis (hence a mismatch between theory and your experiments).
> >
> > I'll keep my current score (leaning towards acceptance).

---

### Official Review · Reviewer_UMDH · 2026-03-10

**Soundness:** 3
**Presentation:** 3
**Significance:** 3
**Originality:** 3
**Overall Recommendation:** 4
**Confidence:** 4

**Summary:**

This paper studies Learning from Language Feedback (LLF), where an agent does not observe rewards directly, but instead receives textual feedback generated by an unknown ground-truth hypothesis. The paper formalizes this setting using a hypothesis class, a verifier loss for checking consistency between feedback and candidate hypotheses, and a new complexity notion called the transfer eluder dimension, which quantifies how informative the feedback is for learning the hidden reward. The authors then propose a UCB-style algorithm, HELiX, and prove sublinear regret guarantees under assumptions such as access to the reward mapping and an unbiased verifier/feedback structure. The paper also includes small-scale experiments on Wordle, Battleship, and Minesweeper.

**Compliance With Llm Reviewing Policy:**

Affirmed.

**Final Justification:**

The paper builds a framework for learning from language feedback. Strengths and weaknesses were summarized previously.

Due to the further explanations and commitments of the authors towards strengthening the paper, with a particular focus on reducing overclaims and generalizing the model, I increased my final score.

**Key Questions For Authors:**

- How essential is the assumption that the reward mapping η↦r_ηis known? Do any of the results extend to settings where the reward semantics are only partially specified?
- How robust are the guarantees to verifier misspecification? Assumption 3 seems quite strong, especially for realistic LLM-based feedback where the verifier may be noisy or biased.
- How should I interpret the transfer eluder dimension? Since the paper notes that it is not a lower bound in general, is it meant as the right notion of LLF hardness, or mainly as an analysis tool for upper bounds?
- Why are the empirical comparisons mostly against a greedy baseline? It would help to understand how the method compares to stronger exploration-based or hypothesis-tracking baselines.

**Limitations:**

- How essential is the assumption that the reward mapping $\eta\to r_\eta$ is known? Do any of the results extend to settings where the reward semantics are only partially specified?
- How robust are the guarantees to verifier misspecification? Assumption 3 seems quite strong, especially for realistic LLM-based feedback where the verifier may be noisy or biased.
- How should I interpret the transfer eluder dimension? Since the paper notes that it is not a lower bound in general, is it meant as the right notion of LLF hardness, or mainly as an analysis tool for upper bounds?
- Why are the empirical comparisons mostly against a greedy baseline? It would help to understand how the method compares to stronger exploration-based or hypothesis-tracking baselines.

**Strengths And Weaknesses:**

*Strengths*
- The authors take the problem of learning from language feedback, which has mostly been studied empirically, and define a theoretical model that captures the underlying difficulty or complexity of learning the latent hypothesis.
- The transfer eluder dimension, introduced as a new complexity measure, is intended to capture when language feedback provides more information than scalar reward, and the examples do a good job of illustrating why that matters.
- In addition to the empirical experiments, the paper explicitly provides an algorithm, HELiX, and proves regret guarantees.

*Weaknesses*
- The paper’s framing feels somewhat broader than what is actually established. In my view, the main contribution is a verifier-based formalization of LLF with guarantees under strong assumptions, rather than a general theory of learning from language.
- In addition, the assumptions support the theoretical results by removing some of the more technically challenging parts of the problem. In particular, the agent is assumed to know the reward mapping and to have access to a verifier that can meaningfully check whether feedback matches a hypothesis. That makes the analysis cleaner, but also makes the setting feel much easier than realistic LLM-based feedback.
- I do not think the transfer eluder dimension is fully convincing as the main hardness notion. The paper itself points out cases that are easy even when this quantity is infinite, so I came away feeling that it is useful for the analysis, but not yet the right notion of problem difficulty.
- The experiments are small, mostly benchmark-style demonstrations, and the baseline comparisons are not especially strong. So, for me, the empirical section mainly shows that the method can work in simple settings, not that it has broad practical significance.

---

> ### Author Rebuttal · Authors · 2026-03-31
>
> Thank you for your insightful questions and feedback.
>
> > The paper’s framing feels somewhat broader than what is actually established. In my view, the main contribution is a verifier-based formalization of LLF with guarantees under strong assumptions, rather than a general theory of learning from language.
>
> We agree that our primary contribution is a rigorous LLF framework under principled assumptions. We do not claim a general assumption-free theory of LLF, but view our work as a first step toward formalizing LLF with assumptions aligned with how natural language is understood in practice. In the introduction, we have emphasized that our framework and results abide by suitable assumptions.
>
> > How essential is the assumption that the reward mapping η↦r_η is known? Do any of the results extend to settings where the reward semantics are only partially specified?
>
> The assumption that the reward mapping is known is standard in theories of bandit with generalization [1-2]. For example, linear bandits assume a known feature map linking $\theta$ to rewards. Without such structure, generalization guarantees are impossible [3].
>
> [1] Chu et al. Contextual Bandits with Linear Payoff Functions. AISTATS 2011.
>
> [2] Grünewälder et al. Regret Bounds for Gaussian Process Bandit Problems. AISTATS 2010.
>
> [3] Bogunovic, I. & Andreas, K. "Misspecified gaussian process bandit optimization." NeurIPS 2021.
>
> > How robust are the guarantees to verifier misspecification? Assumption 3 seems quite strong, especially for realistic LLM-based feedback where the verifier may be noisy or biased.
>
> Our guarantees are robust to moderate verifier misspecification. Assumption 3 only requires unbiasedness in expectation over the feedback distribution, not correctness on every instance. Thus, an LLM-based verifier may make occasional errors as long as these are not systematically biased. More generally, if we allow a $\Delta$-approximate verifier, i.e., $E_{O\sim f_\eta(a)}[\ell(a,O,\eta)] \le \ell_\eta^{\min}(a) + \Delta$), then the regret bound acquires an additional linear bias term of order $O(\sqrt{\Delta})$. Hence, $\Delta$ quantifies the impact of verifier error on performance. We will clarify this in the paper.
>
> > How should I interpret the transfer eluder dimension? Since the paper notes that it is not a lower bound in general, is it meant as the right notion of LLF hardness, or mainly as an analysis tool for upper bounds?
>
> We agree that the transfer eluder dimension does not fully capture the hardness of LLF problems. We discuss this limitation in Section 7. However, the counter example, where $\dim_{TE}$ is infinite while learning is easy, arises in a special ordinal-feedback setting that conveys no reward magnitude. A promising direction is an information-theoretic complexity measure capturing the mutual information between optimal actions and feedback [4]. We will clarify this in Section 7.
>
> [4] Lattimore, T. & Szepesvári, C. An Information-Theoretic Approach to Minimax Regret in Partial Monitoring. COLT 2019.
>
> > For me, the empirical section mainly shows that the method can work in simple settings, not that it has broad practical significance.
>
> We agree that our experiments are conducted in controlled settings and are not intended to demonstrate SOTA performance on complex benchmarks. Instead, they serve as a proof of concept that the proposed framework can be concretely instantiated and aligns with our theory. The primary focus of our paper is a principled theoretical framework to analyze LLF and identify key quantities required for provable learning. This goes beyond task-specific empirical gains toward a principled understanding of when and why language feedback enables learning. While the current experiments are limited to three domains, the underlying framework and design principles are broadly applicable to other tasks. We view our empirical results as a first step that validates feasibility. Scaling to more realistic settings is an important direction for future work.
>
> > It would help to understand how the method compares to stronger exploration-based or hypothesis-tracking baselines.
>
> Our baseline reflects standard practice: given a task, solution, and feedback, the model directly generates an improved solution [5]. It involves no explicit exploration, hence called “CoT" (or "greedy”). HELiX can be layered on top of such approach as an orthogonal module. We ran experiments to compare to a naive exploration-based baseline, $\epsilon$-greedy, by asking the CoT agent to also produce a random action, selected with probability $\epsilon$. We use a linear $\epsilon$ schedule from 0.4 to 0. Results are here: https://anonymous.4open.science/r/LLF-images-2671/. We're happy to add them to the paper. An exploratory action could be worse in minesweeper since the game terminates once you touch a mine - so the cost for random exploration is high.
>
> [5] Yao et al. ReAct: Synergizing Reasoning and Acting in Language Models. ICLR 2023.

---

> > ### Author Rebuttal · Reviewer_UMDH · 2026-04-03
> >
> > Thank you for your detailed answers. I have one follow-up question.
> >
> > The regret analysis relies on the assumption that the reward mapping
> > $\eta \mapsto r_\eta$ is known, which you note is standard in bandit settings.
> > However, in the LLF setting—especially with LLM-based implementations—the reward
> > mapping is typically instantiated via a learned or prompted model, and may therefore
> > be misspecified rather than exact.
> >
> > In this context, could your analysis be extended to a setting where the agent only
> > has access to an approximate reward mapping $\tilde r_\eta$, satisfying, e.g.,
> > $\sup_{a \in \mathcal{A},\eta \in \mathcal{H}}
> > \left| \tilde r_\eta(a) - r_\eta(a) \right| \le \delta$?
> > If so, do the regret guarantees degrade gracefully in $\delta$ (for instance, via an
> > additive or multiplicative term)? Clarifying this would help assess how robust the
> > theory is to reward misspecification in practical LLM-based settings.

---

> > > ### Author Response · Authors · 2026-04-05
> > >
> > > Thank you for the follow-up question. Your proposed relaxation is intuitive and would be valuable to add to the paper given how it ties to the intuition for LLM-based LLF agents. The regret analysis can be extended to accommodate an approximate reward mapping $\tilde{r}\_\eta$ satisfying $\sup\_{a,\eta}|r\_\eta(a) - \tilde{r}\_\eta(a)| \le \delta$.  In that case, the regret decomposition in Proposition 5 will accrue an additional approximation error $2\delta$ at each step, accumulating to an additive order $O(\delta T)$ term on the regret upper bound. Another subtlety this creates is in the exploitation step, as an approximate minimax criteria no longer guarantees zero regret every step onwards. If we modify HELiX to take the exploitation step when the approximate minimax value is zero, i.e., $\tilde{r}\_{\eta\_p}(\pi\_{\eta\_p}) - \tilde{r}\_{\eta\_p}(\pi\_p) = 0$, then the regret bound accrues an additional additive term $2\delta T$ (by modifying Lemma 5). Putting these together, the regret guarantee does degrade gracefully via an additive $O(\delta T)$ term. We will add this discussion and detailed results to the paper.
> > >
> > > We hope this clarification is helpful in assessing the robustness and practical relevance of our framework in LLM-based LLF settings.

---

### Official Review · Reviewer_XGGv · 2026-03-12

**Soundness:** 4
**Presentation:** 3
**Significance:** 3
**Originality:** 4
**Overall Recommendation:** 4
**Confidence:** 3

**Summary:**

This paper introduces a theoretical framework that formalizes the problem of Learning from Language Feedback (LLF) and gives a no-regret algorithm that provably solves LLF through sequential interactions.

The LLF framework is based on assuming the language feedbcak can be processed by a hypothesis verifier, and the definition of transfer eluder dimension, which captures the information contained in language feedback. Through the proposed LLF framework, it can be shown that learning from language feedback is at least as efficient as learning from pure reward, as long as the language feedback is discriminative, by showing that under certaina assumptions, the transfer eluder dimension is always upper bounded by the eluder dimension for learning directly from reward.

The authors then proposed an algorithm HELiX, that iteratively eliminates the hypotheses "too inconsistent" with historical observations, and optimistically chooses a policy that maximizes the reward over all consistent hypothesis until there exists a policy maximizing reward for all remaining hypotheses. It is shown that HELiX has a regret of $\widetilde{O}(T^{3/4}\sqrt{dim_{TE}})$ where $dim_{TE}$ is the transfer eluder dimension, and can be improved to $\widetilde{O}(\sqrt{T})$ (dependence on $dim_{TE}$ omitted) with more structural knowledge on loss structure. It is further discussed how to approximately implement HELiX with LLMs, and experimental validations show that HELiX indeed yields higher cumulative reward than conventional CoT in Battleship and Minesweeper environments, and the exploitation step indeed further boosts the performance.

**Compliance With Llm Reviewing Policy:**

Affirmed.

**Final Justification:**

All of my concerns are addressed and I will maintain my positive score.

**Key Questions For Authors:**

I have the following questions:
1. Is it necessary to define the hypothesis as a text sequence? What is the highest level of abstraction of the hypothesis class for the results to hold? For example, must the hypothesis be discrete? Can it be the reward itself?
2. What is the definition of "consistent" in Assumption 2? If a hypothesis is "inconsistent" with the language feedbcak, does a larger loss mean the hypothesis is "more inconsistent"?
3. In Definition 2, why is $\epsilon^2$ used for loss difference, but $\epsilon$ used for reward differences? How is it technically related to the proof of the theorems? Would similar results hold if transfer dependency is not defined in this way? When $\epsilon$ gets larger, is $\epsilon$-transfer dependency a weaker or stronger notion? How to connect this with Definition 5?
4. Can you provide concrete examples for transfer eluder dimension, why does it capture information in the language feedback, and give a high-level categorization on how/when can we compute this transfer eluder dimension?
5. How does the exploitation step affect the regret bound in Theorem 1?
6. How frequently is the exploitation step used in the experiments? Figure 2 uses the number of exploration step as x-axis, what is an exploration step for CoT? Additionally, how does HELiX without exploitation step compare with HELiX if we count the total steps (both exploration and exploitation?) How does HELiX compare with CoT (or simple extensions of CoT) when the same number of tokens is used?

**Limitations:**

Yes.

**Strengths And Weaknesses:**

Strengths:

1. The paper is generally well-written. The framework of LLF is novel and interesting both theoretically and practically. Most design choices and assumptions made seem reasonable to me.
2. Through the definition of transfer eluder dimension, it is shown through Example 1 that language feedback can indeed reduce learning complexity. Furthermore, it is shown in Proposition 1 that when feedback is discriminative, LLF is at least as efficient as learning from reward. These results provide interesting theoretical insights for the underlying intuitions.
3. An algorithm HELiX is proposed, and the regret bound of HELiX is given. Experimental adaptation of HELiX is discussed and experiments are provided showcasing HELiX outperforms CoT and the exploitation step is indeed helpful. This motivates future research on how to structurally improve CoT with theoretical insights.

Weaknesses:

1. While the transfer eluder dimension captures the complexity of the LLF problem, it is stated that computing or estimating the transfer eluder dimension is hard, and no further discussion on the class of problems in which transfer eluder dimension can be computed or estimated other than one example.
2. Minor: The experimental adaptation of HELiX seems a bit too far away from the algorithm itself. The ablation study on different possible choices of approximating each step is relatively limited for hypothesis consistency verification and minimax exploitation.

---

> ### Author Rebuttal · Authors · 2026-03-31
>
> Thank you for the positive review and constructive feedback.
>
> > Can you provide concrete examples for transfer eluder dimension, why does it capture information in the language feedback, and give a high-level categorization on how/when can we compute this transfer eluder dimension?
>
> In addition to Example 1, we provide another concrete example in Appendix D, Example 2. More generally, computing the transfer eluder dimension requires specifying the hypothesis verifier loss, reward mapping, and feedback distribution (essentially, the feedback mapping). This typically entails application-dependent assumptions about the LLM and feedback oracle. We hope that the concrete examples here provide insights into how these components (e.g. information in feedback) affect learning complexity through transfer eluder dimension.
>
> > The experimental adaptation of HELiX seems a bit too far away from the algorithm itself. The ablation study on different possible choices of approximating each step is relatively limited for hypothesis consistency verification and minimax exploitation.
>
> We designed the empirical HELiX implementation to follow the theoretical algorithm line by line. However, each step requires LLM-based instantiations, making approximations unavoidable. We validated that each component is reasonably approximated: prompt tuning ensures robust hypothesis verification, Appendix F.7 evaluates the empirical validity of Assumption 3, and Appendix F.3 (Figure 5) contains additional ablations of design choices (there greedy = CoT). We will highlight these in the main paper.
>
> > Is it necessary to define the hypothesis as a text sequence? What is the highest level of abstraction of the hypothesis class for the results to hold? For example, must the hypothesis be discrete? Can it be the reward itself?
>
> The hypothesis space is modeled as a subspace of text to reflect autoregressive LLM outputs, but this is not essential. Our formulation and results extend to general hypothesis spaces, whether discrete or continuous. If the reward mapping is the identity, a hypothesis reduces to a reward function. More broadly, as noted in Section 7, different hypothesis designs correspond to different learning targets, affecting both behavior and complexity; characterizing these tradeoffs is an important theoretical and empirical direction.
>
> > If a hypothesis is "inconsistent" with the language feedback, does a larger loss mean the hypothesis is "more inconsistent"?
>
> Yes, the intuition is correct. One goal of an effective agent is to identify hypotheses achieving low average hypothesis verifier loss.
>
> > Why is \epsilon^2 used for loss difference, but \epsilon used for reward differences? How is it technically related to the proof of the theorems? Would similar results hold if transfer dependency is not defined in this way?
>
> The technical reason is a clean discriminative feedback condition in Proposition 1. To relate transfer eluder dimension to the usual eluder dimension with the same scaling, we replace the role of cumulative squared reward error by the cumulative hypothesis verifier loss. When the loss is taken to be a squared prediction loss, the $\epsilon^2$ scaling matches the rate in the definition of eluder dimension. Using a different scaling would still yield qualitatively similar results, but with a rescaling in the comparison, say $\dim_{TE}(H,C_F\ell,\epsilon^2) \le \dim_E(R_H,\epsilon)$.
>
> > When \epsilon gets larger, is \epsilon-transfer dependency a weaker or stronger notion? How to connect this with Definition 5?
>
> As $\epsilon$ increases, $\epsilon$-transfer dependence becomes weaker: larger $\epsilon$ allows more tolerance, so fewer actions remain independent. With Definition 5, we connect transfer eluder dimension to eluder dimension through Proposition 1. The $\epsilon$-eluder dimension is nonincreasing with $\epsilon$, giving a tighter upper bound on the $\epsilon$-transfer eluder dimension.
>
> > How does the exploitation step affect the regret bound in Theorem 1?
>
> The exploitation step does not affect the regret bound. It is an early stopping criteria necessarily reducing regret of HELiX without this step. We explained this point in Lines 990-993 in the Appendix.
>
> > Figure 2 uses the number of exploration steps as x-axis, what is an exploration step for CoT? Additionally, how does HELiX without exploitation step compare with HELiX if we count the total steps (both exploration and exploitation?)
>
> The x-axis stands for the total number of steps (both exploration and exploitation). Thanks for pointing out the confusion in the axis labeling. We will change “exploration steps” to “interaction steps”.
>
> > How does HELiX compare with CoT (or simple extensions of CoT) when the same number of tokens is used?
>
> It is not easy to fix the total number of tokens used for HELiX and CoT due to the stochasticity of text generation. Instead, we provide a study on computational cost in Appendix F.8.

---

> > ### Author Rebuttal · Reviewer_XGGv · 2026-04-02
> >
> > Thank the authors for the detailed response, all of my concerns are addressed and I will maintain my positive score.

---

### Official Review · Reviewer_2TgZ · 2026-03-13

**Soundness:** 4
**Presentation:** 4
**Significance:** 4
**Originality:** 3
**Overall Recommendation:** 5
**Confidence:** 3

**Summary:**

This paper proposes a theoretical framework to formalize the problem of learning from language feedback. Specifically, this notion of how informative a piece of verifier feedback is is captured by a new quantity called the transfer eluder dimension (based on the original eluder dimension). This quantity is used to upper bound the regret of the HELiX algorithm proposed by the authors, which is proven to be sublinear in the transfer eluder dimension. Experiments on several natural language tasks are provided.

**Compliance With Llm Reviewing Policy:**

Affirmed.

**Final Justification:**

All of my concerns have been addressed, and I do believe this paper deserves an accept at the conference. Rigorous theory on LLMs is hard to come across these days, and learning from language feedback setting is an important one to understand (there is plenty of room for empirical improvement as well).

**Key Questions For Authors:**

- Is there any way to establish that practical language tasks can have small transfer eluder dimension?
- Have the authors tried their practical algorithm on other tasks besides the three mentioned to see if the gap between HELiX and CoT remains statistically significant?

**Limitations:**

Yes.

**Strengths And Weaknesses:**

Strengths:
- This work captures the intuition that we should somehow train models to give more informative feedback, rather than the widely adopted setting of (for instance in RLVR) a single bit of outcome reward.
- The formalization in terms of regret minimization is quite a clean one, with nontrivial guarantees as stated in Theorem 1.
- Results on synthetic natural language tasks support the overall argument.

Weaknesses:
- Somehow, the assumptions feel slightly strong. In practice we have access to noisy versions of the quantities (say, via LLM-as-a-judge). It would be interesting to understanding how that would affect the analyses in the paper.
- As acknowledged by the authors, the transfer eluder dimension may not be the right notion of the regret lower bound, and it would be interesting to see further work in this direction.

---

> ### Author Rebuttal · Authors · 2026-03-31
>
> Thank you for the positive review and constructive feedback.
> > Somehow, the assumptions feel slightly strong. In practice we have access to noisy versions of the quantities (say, via LLM-as-a-judge). It would be interesting to understand how that would affect the analyses in the paper.
>
> Our verifier assumption can be naturally extended to allow noise and small approximation error in the verifier loss. In Assumption 3, unbiasedness is defined for the average loss over the distribution of observed feedback. In particular, an LLM-based hypothesis verifier may occasionally misjudge individual feedback instances, but as long as its errors are not systematically biased—i.e., it is correct on average over the distribution of feedback—it satisfies the assumption. If we assume a $\Delta$-approximately correct verifier to model potential LLM mistakes (formalized by modifying Assumption 3 to $E_{O\sim f_\eta(a)}[\ell(a,O,\eta)] \le \ell_\eta^{\min}(a) + \Delta$), then the regret bound acquires an additional linear bias term of order $O(\sqrt{\Delta})$. In other words, $\Delta$ controls how much the regret guarantee degrades under verifier error. We will add this discussion to the paper.
>
> > Is there any way to establish that practical language tasks can have small transfer eluder dimension?
>
> We provide a concrete example in Example 1 (lines 212-245) demonstrating how transfer eluder dimension varies as the information in feedback.
>
> > Have the authors tried their practical algorithm on other tasks besides the three mentioned to see if the gap between HELiX and CoT remains statistically significant?
>
> The goal of these experiments is mainly to provide a proof-of-concept that the proposed framework can be instantiated concretely and aligns with theory. The main focus of our paper is on offering a theoretical framework for people to mathematically analyze the paradigm of learning from language feedback and what formal quantities are needed to make an algorithm “provably” learning. This goes beyond task-specific empirical gains, but toward a principled understanding of when and why language feedback enables learning. While the current experiments are limited to three domains, the underlying framework and design principles are broadly applicable to other tasks too. We view our empirical results as a first step that validates feasibility and effectiveness of LLF algorithms designed with theoretical principles. We believe scaling to more realistic settings is an important direction for future work.

---

> > ### Author Rebuttal · Reviewer_2TgZ · 2026-04-02
> >
> > Thanks to the authors for their response. All of my concerns have been addressed, and I do believe this paper deserves an accept at the conference.
> >
> > Rigorous theory on LLMs is hard to come across these days, and learning from language feedback setting is an important one to understand (there is plenty of room for empirical improvement as well).

---

### Decision · Program_Chairs · 2026-04-30

**Decision:**

Accept (regular)

**Comment:**

This is a paper that theoretically formalizes the problem of learning from language feedback. While some assumptions can be questioned, it is still a nice initial attempt that will attract a sufficient amount audience from ICML. Please take into account the key discussion points in the final version including reducing overclaims and generalizing the model.